# Reconstructing the Mind's Eye: fMRI-to-Image with Contrastive Learning and Diffusion Priors

**Paul S. Scotti**[*,1,2], **Atmadeep Banerjee**[*,2], **Jimmie Goode**[†,2], **Stepan Shabalin**[2], **Alex Nguyen**[1],
**Ethan Cohen**[3], **Aidan J. Dempster**[4], **Nathalie Verlinde**[1], **Elad Yundler**[5], **David Weisberg**[1,2],
**Kenneth A. Norman**[‡,1], and **Tanishq Mathew Abraham**[‡,2,6,7]

[1]Princeton Neuroscience Institute
[2]Medical AI Research Center (MedARC)
[3]Ecole Normale Supérieure, PSL University
[4]University of Toronto
[5]Hebrew University of Jerusalem
[6]EleutherAI
[7]Stability AI
*Project Page:* `https://medarc.ai/mindeye/`

## Abstract

We present MindEye, a novel fMRI-to-image approach to retrieve and reconstruct viewed images from brain activity. Our model comprises two parallel submodules that are specialized for retrieval (using contrastive learning) and reconstruction (using a diffusion prior). MindEye can map fMRI brain activity to any high dimensional multimodal latent space, like CLIP image space, enabling image reconstruction using generative models that accept embeddings from this latent space. We comprehensively compare our approach with other existing methods, using both qualitative side-by-side comparisons and quantitative evaluations, and show that MindEye achieves state-of-the-art performance in both reconstruction and retrieval tasks. In particular, MindEye can retrieve the exact original image even among highly similar candidates, indicating that its brain embeddings retain fine-grained image-specific information. This allows us to accurately retrieve images even from large-scale databases like LAION-5B. We demonstrate through ablations that MindEye's performance improvements over previous methods result from specialized submodules for retrieval and reconstruction, improved training techniques, and training models with orders of magnitude more parameters. Furthermore, we show that MindEye can better preserve low-level image features in the reconstructions by using img2img, with outputs from a separate autoencoder. All code is available on GitHub.

## 1 Introduction

The problem of decoding environmental inputs and cognitive states from brain activity is fundamental to the field of neuroscience, where improved computational approaches allow for further understanding of brain mechanisms [1]. A neuroimaging methodology that has seen significant success in this domain is functional magnetic resonance imaging (fMRI), where neural activity is measured by detecting changes in blood oxygenation. fMRI decoding is already being used in

---

[*]Equal contributions.

[†]Core contribution.

[‡]Joint senior authors.

37th Conference on Neural Information Processing Systems (NeurIPS 2023).

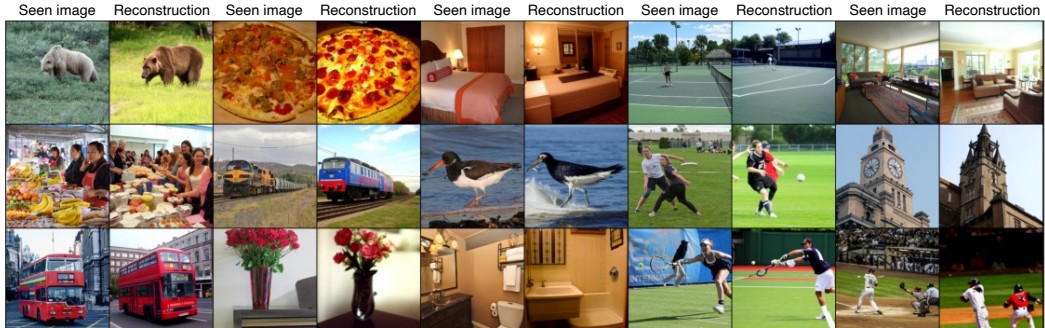

Figure 1: Example images reconstructed from human brain activity corresponding to passive viewing of natural scenes. Reconstructions depict outputs from Versatile Diffusion [6] given CLIP fMRI embeddings generated by MindEye for Subject 1. See Figure 4 and Appendix A.4 for more samples.

real-time clinical domains [2] and has potential for novel mind reading applications in brain-computer interfaces. Previous works mapped fMRI activity to the embeddings of image generation models via relatively simple mappings, usually ridge regression [3–5]. Here we propose MindEye, a novel approach that involves mapping via large-scale multilayer perceptrons (MLPs), contrastive learning, and diffusion models to achieve state-of-the-art image reconstruction. See Figure 1 for select samples of reconstructions.[1]

MindEye learns to map flattened spatial patterns of fMRI activity across voxels (3-dimensional cubes of cortical tissue) to the image embedding latent space of a pretrained CLIP [7] model. MindEye has an MLP backbone and 2 specialized submodules for retrieval and reconstruction. The retrieval submodule is contrastively trained and produces "disjointed CLIP fMRI" embeddings that have high cosine similarity with the corresponding image embeddings but differ in magnitude. To reconstruct images, we train a diffusion prior [8] from scratch to take in the outputs from the MLP backbone and produce aligned embeddings suitable as inputs to any pretrained image generation model that accepts CLIP image embeddings. In order to ensure that our reconstructions also match the original images' low-level features (e.g., color, texture, spatial position), we train a separate encoder that directly maps voxels to the embedding space of Stable Diffusion's [9] variational autoencoder (VAE), obtaining blurry image reconstructions that lack high-level semantic content but perform state-of-the-art on low-level image metrics. Combining the high-level "semantic" pipeline with the low-level "perceptual" pipeline in an img2img [10] setting allows MindEye to output state-of-the-art reconstructions across both low- and high-level image metrics.

In addition to image reconstruction metrics, our disjointed CLIP fMRI embeddings attain state-of-the-art performance on image retrieval and brain retrieval metrics. Image retrieval refers to finding the original seen image out of a pool of other images given a brain sample, while brain retrieval refers to finding the brain sample given an image. MindEye finds exact (top-1) matches in the pool of NSD test samples with >90% accuracy for both image and brain retrieval, outperforming previous state-of-the-art [11, 4] which showed <50% retrieval accuracies. These results demonstrate that MindEye brain embeddings possess fine-grained exemplar-level signal.

Our main findings are: (1) Specialized submodules for retrieval (using contrastive learning) and reconstruction (using a diffusion prior) enable a single model to achieve state-of-the-art results across both tasks even though the objectives exhibit a tradeoff. (2) Mapping to a deep MLP with a parameter count orders of magnitude higher than previous methods does not produce overfitting and instead directly benefits model performance. (3) A novel bidirectional version of mixup contrastive data augmentation further improves model performance in this low-sample setting. (4) State-of-the-art reconstructions for low-level image metrics can be obtained by independently mapping to Stable Diffusion's VAE latent space. (5) fMRI-to-image retrieval can find the exact original image even among highly similar candidates, suggesting that fine-grained image-specific information is contained in brain embeddings, thus allowing retrieval to be scaled up to large-scale databases like LAION-5B to output images without generative models.

---

[1]Images containing each subject's 982 test set reconstructions and retrievals are available on GitHub.

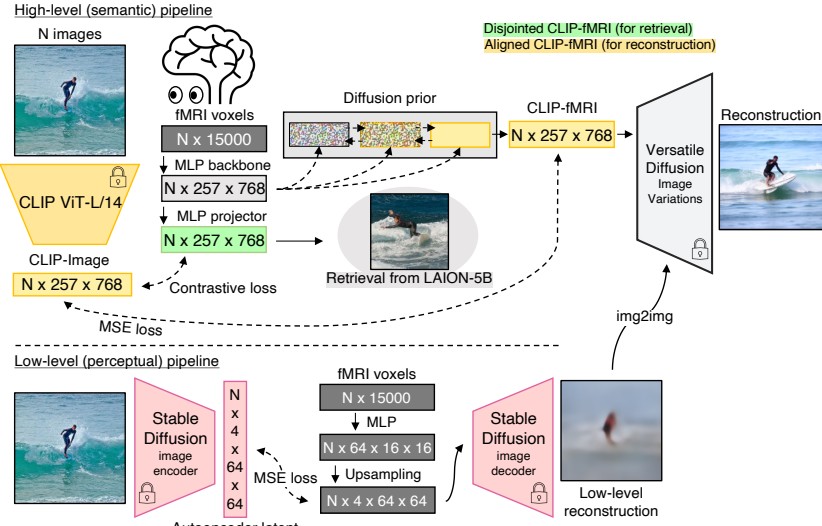

Figure 2: MindEye overall schematic. A high-level "semantic" pipeline maps voxels to CLIP embeddings for image reconstruction (outputs from a diffusion prior feed through generative models like Versatile Diffusion) or retrieval tasks (such as K-nearest neighbor querying of brain embeddings to the CLIP embeddings of LAION-5B images). A low-level "perceptual" pipeline maps voxels to the variational autoencoder used by Stable Diffusion to obtain blurry reconstructions, which are used as the initialization for subsequent diffusion-based image generation. The contrastive loss for the low-level pipeline is omitted for simplicity; see Appendix A.3.2 for details.

## 2 MindEye

MindEye consists of two pipelines (see Figure 2), a high-level (semantic) pipeline where fMRI voxels are mapped to the CLIP ViT-L/14 image space and a low-level (perceptual) pipeline where the voxels are mapped to the image embedding space of a VAE. Both pipelines follow a common structure: a residual MLP backbone followed by two task-specific submodules. For the high-level pipeline the submodules are an MLP projector and a diffusion prior. For the low-level pipeline the submodules are an MLP projector and a CNN decoder that performs 4x upsampling. For both pipelines we observe that training the projector submodule with a contrastive loss and the second submodule with mean squared error (MSE) loss gives the best performance.

### 2.1 High-Level (Semantic) Pipeline

The high-level pipeline is the core of MindEye as it maps voxels to CLIP image space to be fed through pretrained image generation models. We refer to it as a "high-level" pipeline because CLIP embeddings are inherently more semantic than perceptual, given that CLIP image encoders were trained to maximize similarity with text captions (low-level features like color and object location are not typically preserved in these captions). MindEye can be used without the low-level pipeline, which simply aids to better preserve low-level image features during reconstruction.

The MLP backbone for our high-level pipeline maps flattened voxels to an intermediate space of size $257 \times 768$, corresponding to the last hidden layer of CLIP ViT/L-14 (see Appendix 1 for PyTorch model code). The backbone consists of a linear layer followed by 4 residual blocks and a final linear projector. The embeddings from the backbone are fed to an MLP projector and a diffusion prior in parallel. The whole model is trained end-to-end with the prior getting an MSE loss and the projector getting a bidirectional CLIP loss. The projector outputs can be used for retrieval tasks and the diffusion prior outputs can be used by generative models to reconstruct images.

**Contrastive Learning:** Contrastive learning is an effective method for learning representations across modalities by maximizing cosine similarity for positive pairs while minimizing similarity for negative pairs. Previous work has shown the potential benefits of using contrastive learning alongside neural data [12, 13]. CLIP [7] is an example of a multimodal contrastive model that maps

images and text captions to a shared embedding space. MindEye is trained to introduce fMRI as an additional modality to the embedding space of a pretrained CLIP model, keeping the CLIP image space frozen as done with locked-image text tuning (LiT) [14]. We use the CLIP loss [7] as our contrastive objective. This loss is bidirectional and helps improve both image and brain retrieval.

Recent work [15–18] has explored novel data augmentation techniques that offer several benefits like improving performance, increasing robustness, and reducing training data requirements. Mixup [15] is one such technique that trains models on synthetic data created through convex combinations of two datapoint-label pairs [19]. Kim et al. [20] introduce MixCo, an extension of mixup that uses the InfoNCE loss, and show that MixCo improves classification performance in a semi-supervised setting. Based on the same principle, we modify the bidirectional CLIP loss to use MixCo. While Kim et al. [20] observed that MixCo gives largest performance benefit for smaller models, we observe that it also helps large models in low data regimes.

To combine MixCo with CLIP loss, we mix voxels using a factor $\lambda$ sampled from the Beta distribution with $\alpha = \beta = 0.15$.

$$x_{\mathrm{mix}_{i,k_i}} = \lambda_i \cdot x_i + (1 - \lambda_i) \cdot x_{k_i}, \quad p_i^* = f(x_{\mathrm{mix}_{i,k_i}}), \quad p_i = f(x_i), \quad t_i = \mathrm{CLIP}_{\mathrm{Image}}(y_i) \quad (1)$$

Here, $x_i$ and $y_i$ represent the $i$-th fMRI sample and image respectively. $k_i \in [1, N]$ is an arbitrary mixing index for the $i$-th datapoint and $f$ represents the combined MLP and projector. $p^*$, $p$ and $t$ are L2-normalized. The CLIP loss with MixCo is defined as:

$$
\begin{aligned}
\mathcal{L}_{\mathrm{BiMixCo}} = -\sum_{i=1}^{N} & \left[ \lambda_i \cdot \log \left( \frac{\exp\left(\frac{p_i^* \cdot t_i}{\tau}\right)}{\sum_{m=1}^{N} \exp\left(\frac{p_i^* \cdot t_m}{\tau}\right)} \right) + (1 - \lambda_i) \cdot \log \left( \frac{\exp\left(\frac{p_i^* \cdot t_{k_i}}{\tau}\right)}{\sum_{m=1}^{N} \exp\left(\frac{p_i^* \cdot t_m}{\tau}\right)} \right) \right] \\
-\sum_{j=1}^{N} & \left[ \lambda_j \cdot \log \left( \frac{\exp\left(\frac{p_j^* \cdot t_j}{\tau}\right)}{\sum_{m=1}^{N} \exp\left(\frac{p_m^* \cdot t_j}{\tau}\right)} \right) + \sum_{\{l|k_l=j\}} (1 - \lambda_l) \cdot \log \left( \frac{\exp\left(\frac{p_l^* \cdot t_j}{\tau}\right)}{\sum_{m=1}^{N} \exp\left(\frac{p_m^* \cdot t_j}{\tau}\right)} \right) \right]
\end{aligned}
$$
$$(2)$$

We term this bidirectional loss as BiMixCo. Here $\tau$ is a temperature hyperparameter, and $N$ is the batch size.

Recent works [21, 22] have shown that stopping mixup augmentation after a certain number of epochs leads to better classification performance. As per these findings, we stop using mixup and switch from a hard contrastive loss to a soft contrastive loss one-third of the way through training. This improves our reconstructions without harming our retrieval performance (see Table 4). BiMixCo gives the highest retrieval performance but slightly hurts reconstructions (Table 4) likely due to how the reconstruction task needs absolute targets for the mixed inputs (which we generate by mixing the original targets in the same ratio as the mixup inputs), causing a slight shift in the distribution of target embeddings. Our final schedule combines BiMixCo and soft contrastive loss to strike the best balance between retrieval and reconstruction performance in a single model.

Our soft contrastive loss is inspired by knowledge distillation [23], where the authors argue that the softmax probability distribution produced by a powerful teacher model acts as a better teaching signal for a student than hard labels. To generate the soft labels we take the dot product of CLIP image embeddings in a batch with themselves. The loss (with bidirectional component omitted for brevity) is calculated between CLIP-CLIP and Brain-CLIP matrices as:

$$\mathcal{L}_{\mathrm{SoftCLIP}} = -\sum_{i=1}^{N} \sum_{j=1}^{N} \left[ \frac{\exp\left(\frac{t_i \cdot t_j}{\tau}\right)}{\sum_{m=1}^{N} \exp\left(\frac{t_i \cdot t_m}{\tau}\right)} \cdot \log \left( \frac{\exp\left(\frac{p_i \cdot t_j}{\tau}\right)}{\sum_{m=1}^{N} \exp\left(\frac{p_i \cdot t_m}{\tau}\right)} \right) \right] \quad (3)$$

**Diffusion Prior:** Using a diffusion model to align the outputs of a contrastive learning model was inspired by DALL-E 2 [8], where a "diffusion prior" was used to map CLIP text embeddings to CLIP

image space before using an unCLIP decoder to reconstruct images. Here we train our own diffusion prior from scratch to map CLIP fMRI embeddings to CLIP image space, which are then fed into a pretrained Versatile Diffusion model to generate image reconstructions. We modified an open-source implementation of the DALL-E 2 diffusion prior available on GitHub (see Appendix A.3.1). We used the same prior loss as Ramesh et al. [8]. Our total end-to-end loss is defined as:

$$\mathcal{L} = \mathcal{L}_{\text{BiMixCo|SoftCLIP}} + \alpha \cdot \mathcal{L}_{\text{prior}} \qquad (4)$$

We use $\alpha = 0.3$ and switch from BiMixCo to SoftCLIP after one-third of the train cycle. All our models are trained on a single A100 GPU for 240 epochs with a batch size of 32. Despite a high parameter count, MindEye (including both high- and low-level pipelines) can be trained on a single A100 in less than 18 hours. This efficiency is due to the bulk of the parameters stemming from MLPs, which are faster to compute than transformers or CNNs.

The diffusion prior is critical for reconstruction because contrastive learning only incentivizes the CLIP fMRI embeddings to match the vector direction of the associated CLIP image embeddings. This generates disjointed embeddings as observed by Ramesh et al. [8]. Theoretically, multimodal contrastive learning will always produce disjointed embeddings because of the "modality gap" phenomenon whereby encoding modalities into a shared space restricts the effective embedding space to a narrow cone in geometric space [24]. We use pre-trained models that expect CLIP image embeddings as input, thus motivating our training of a diffusion prior to align disjointed embeddings.

To rectify this issue, the diffusion prior learns a distribution of CLIP image embeddings conditioned on CLIP fMRI embeddings. UMAP [25] plots of disjointed CLIP fMRI embeddings next to aligned CLIP fMRI embeddings in Appendix A.5 show how the diffusion prior addresses the disjointed embedding spaces problem. We observe that the prior's role cannot be fulfilled by simply adding MSE loss to the MLP projector in Table 4. This is because there is a tradeoff between reconstruction and retrieval objectives and a model cannot effectively learn a single embedding space that does well on both.

## 2.2 Low-Level (Perceptual) Pipeline

The low-level pipeline maps voxels to the embedding space of Stable Diffusion's VAE. The output of this pipeline can be fed into the VAE decoder to produce blurry image reconstructions that lack high-level semantic content but exhibit state-of-the-art low-level image metrics. We use img2img [10] to improve our final image reconstructions in terms of low-level metrics, with minimal impairment to high-level metrics, such that we start the diffusion process from the noised encodings of our blurry reconstructions rather than pure noise.

The MLP backbone for our low-level pipeline follows the same architecure as our high-level pipeline, except that the final outputs are of size $(16, 16, 64)$. These are upsampled to $(64, 64, 4)$ by a CNN upsampler. An MLP projector projects the backbone outputs to a $512$ dimensional space where an auxiliary contrastive loss is applied. For more information on the low-level pipeline see Appendix A.3.2. See Appendix Figure 7 for example blurry reconstructions and Appendix Table 5 to see the effect of changing img2img strength on subsequent reconstruction metrics.

## 3 Results

For all experiments, we used the Natural Scenes Dataset (NSD) [26], a public fMRI dataset containing the brain responses of human participants passively viewing natural scenes from MS-COCO [27]. By utilizing MS-COCO, this dataset provides measured brain responses to rich naturalistic stimuli, allowing us to study how well low- and high-level image features are reconstructed by MindEye. We used the same standardized train/test splits as other NSD reconstruction papers [3, 4, 28], training subject-specific models for each of 4 participants. We averaged across three same-image repetitions for the test set (leaving 982 test samples) but not the training set (24,980 training samples), similar to Takagi and Nishimoto [3]. For more information on NSD and data preprocessing see Appendix A.2; for single-trial and reduced dataset results see Appendix A.9 and Appendix A.10.

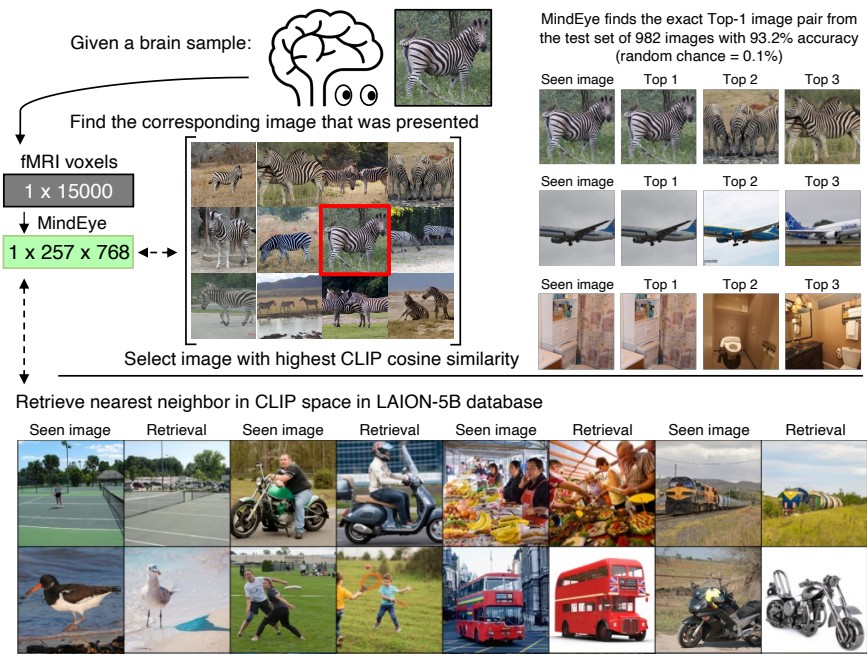

Figure 3: MindEye image retrieval. Given a pool of candidate images, nearest neighbor search in CLIP space enables searching for the original image based on brain activity. Top section depicts how, given 982 test NSD images (many containing very similar looking images, e.g., over a dozen zebras), MindEye top-1 performance is 93.2% for Subject 1. The ability to distinguish among confusable candidates suggests brain embeddings retain fine-grained, image-specific information. Bottom section depicts scaling up to the LAION-5B dataset (see Appendix A.4 for more examples). Even with billions of images, MindEye finds images similar to the original.

## 3.1    Image/Brain Retrieval

Image retrieval evaluations reveal the level of fine-grained image-specific information contained in the predicted brain embeddings. For example, if the model is given a dozen pictures of zebras and the brain sample corresponding to viewing one of those zebras, can the model correctly find the corresponding zebra? If the model can correctly deduce that the brain sample corresponds to an image of a zebra but cannot deduce the specific image amongst various candidates, this would suggest that category-level information but not exemplar-specific information is preserved in the CLIP fMRI embedding. MindEye not only succeeds in this zebra example but also demonstrates 93.2% overall accuracy for Subject 1 in finding the exact original image within the 982 test images (see Figure 3).

Although we use the full test dataset for retrievals in Figure 3, we followed the same procedure as Lin et al. [11] for calculating the retrieval metrics reported in Table 1. Brain retrieval performance was calculated according to the following procedure: for each test image, the image is converted to a CLIP image embedding and we compute the cosine similarity to both its respective ground truth disjointed CLIP fMRI embedding as well as 299 other randomly selected disjointed CLIP fMRI embeddings in the test set. For each test sample, success is determined if the cosine similarity is greatest between the ground truth CLIP embedding and its respective fMRI embedding (aka top-1 retrieval performance, chance=1/300). We average retrieval performance across all test samples and repeat the entire process 30 times to account for the variability in random sampling of batches. For image retrieval, the same procedure is used except image and brain samples are flipped such that the goal is to find the corresponding paired CLIP image embedding out of 300 possible CLIP embeddings in the batch. Lin et al. [11] refer to image retrieval as "forward retrieval" and brain retrieval as "backward retrieval" in their paper.

We can scale up image retrieval using a pool of billions of image candidates. In Figure 3 we show results querying the LAION-5B dataset [29] using our CLIP fMRI embeddings. The final layer CLIP ViT-L/14 embeddings for all 5 billion images are available at knn.laion.ai, and can be queried for K-nearest neighbor lookup via the CLIP Retrieval client [30]. For each test sample, we first retrieve 16 candidate images using this method (using a variant of MindEye that maps voxels to the final

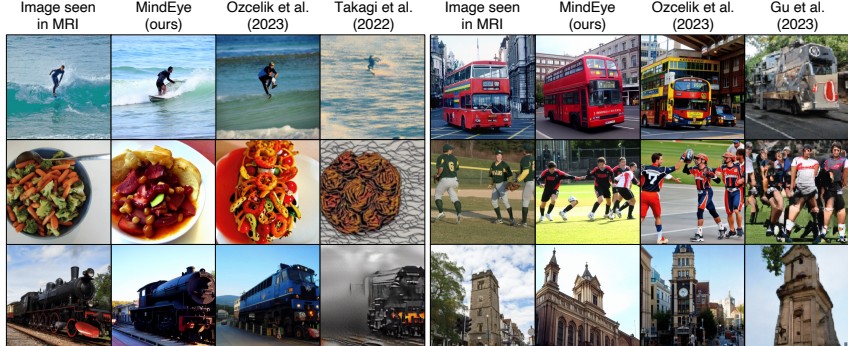

Figure 4: Side-by-side comparison of reconstructions from fMRI-to-Image NSD papers. The same test set was used across papers. All reconstructions come from Subject 1.

layer of CLIP, see Appendix A.7). The best image is then selected based on having the highest CLIP embedding cosine similarity to the CLIP fMRI embedding. This image retrieval approach is especially well-suited for tasks involving fine-grained classification, and can be used as an alternative to image reconstruction without a generative model (evaluations in Table 1).

## 3.2 fMRI-to-Image Reconstruction

The diffusion prior outputs from MindEye are aligned CLIP fMRI embeddings that can be used with any pretrained image generation model that accepts latents from CLIP image space. We evaluate the outputs of MindEye reconstructions across several models including Versatile Diffusion [6], Stable Diffusion (Image Variations) [31], and Lafite [32, 11]. Here we report results from Versatile Diffusion since it yielded the best results, and we report results from the other models in Appendix A.7. We qualitatively compare our reconstructions side-by-side with outputs from other fMRI-to-image reconstruction models in Figure 4 and quantitatively compare against other models in Table 1, demonstrating state-of-the-art MindEye reconstructions.

For each subject, for each test brain sample, we output 16 CLIP image embeddings from MindEye and feed these embeddings through the image variations pipeline of Versatile Diffusion. This produces 16 image reconstructions per brain sample. For our reconstructions we use 20 denoising timesteps with UniPCMultistep noise scheduling [33] and start the denoising process from the noised output of our low-level pipeline (img2img). We then select the best of 16 reconstructions by computing last hidden layer CLIP embeddings and picking the image with the highest cosine similarity to the disjointed CLIP fMRI embedding. This automatic second-order selection was inspired by DALL-E 2 [8], which used a similar process of selecting the best of 2 generated samples.

Two-way identification refers to percent correct across comparisons gauging if the original image embedding is more similar to its paired brain embedding or a randomly selected brain embedding. Comparison was performed for AlexNet [37] (second and fifth layers), InceptionV3 [38] (last pooling layer), and CLIP (final layer of ViT-L/14). We use the same settings as Ozcelik and VanRullen [4] for our metrics. For more details refer to Appendix A.6.

## 3.3 Ablations

In this subsection we try to explain where MindEye performance improvements come from through ablations. To study the effects of architectural changes and training strategies we train only the retrieval pipeline (no diffusion prior) for 120 epochs with batch size 300. All models in this section are trained on Subject 1. Table entries with * correspond to the final version of MindEye's settings.

**Architectural Improvements:** To study the effect of model depth and parameter count we train multiple MLPs of various sizes (Table 2). Among models that map to the last hidden layer of CLIP ViT-L/14, we observe a clear trend of increased performance with added residual blocks. For 2 blocks, the effect of skip connections is not too significant but at 4 blocks the model does significantly worse without them, indicating that skip connections are important for training deeper models.

| Method | Low-Level | | | | High-Level | | | | Retrieval | |
|---|---|---|---|---|---|---|---|---|---|---|
| | PixCorr ↑ | SSIM ↑ | Alex(2) ↑ | Alex(5) ↑ | Incep ↑ | CLIP ↑ | Eff ↓ | SwAV ↓ | Image ↑ | Brain ↑ |
| Lin et al. [11] | – | – | – | – | 78.2% | – | – | – | 11.0% | 49.0% |
| Takagi... [3] | – | – | 83.0% | 83.0% | 76.0% | 77.0% | – | – | – | – |
| Gu et al. [28] | .150 | .325 | – | – | – | – | .862 | .465 | – | – |
| Ozcelik... [4] | .254 | **.356** | 94.2% | 96.2% | 87.2% | 91.5% | .775 | .423 | 21.1% | 30.3% |
| MindEye | **.309** | .323 | **94.7**% | **97.8**% | **93.8**% | **94.1**% | **.645** | **.367** | **93.6**% | **90.1**% |
| MindEye (Low-Level) | **.360** | **.479** | 78.1% | 74.8% | 58.7% | 59.2% | 1.00 | .663 | – | – |
| MindEye (High-Level) | .194 | .308 | **91.7**% | **97.4**% | **93.6**% | **94.2**% | **.645** | .369 | **93.6**% | **90.1**% |
| MindEye (LAION) | .130 | .308 | 84.0% | 92.6% | 86.9% | 86.1% | .778 | .477 | – | – |
| Ozcelik... (Low-, S1) | .358 | .437 | **97.7**% | **97.6**% | **77.0**% | **71.1**% | **.906** | **.581** | – | – |
| MindEye (Low-, S1) | **.456** | **.493** | 87.1% | 84.1% | 61.6% | 62.4% | .992 | .638 | – | – |

Table 1: Quantitative comparison of MindEye retrieval and reconstruction performance against other models. Top and middle sections average across the same 4 participants (see Appendix A.8 for individual subject models), except Lin et al. [11] which only analyzed Subject 1. Middle section reflects outputs from only the high- or low-level pipeline, and metrics when evaluating images retrieved from LAION-5B. Bottom section compares our low-level reconstructions to the low-level reconstructions from Ozcelik and VanRullen [4] which only reported metrics for Subject 1. Image retrieval refers to the percent of the time the correct image was retrieved out of 300 candidates, given the associated brain sample (chance=0.3%); vice-versa for brain retrieval. PixCorr=pixelwise correlation between ground truth and reconstructions; SSIM=structural similarity index metric [34]; EfficientNet-B1 ("Eff") [35] and SwAV-ResNet50 ("SwAV") [36] refer to average correlation distance; all other metrics refer to two-way identification (chance = 50%). Missing values are from papers not reporting all metrics or metrics being non-applicable. We followed the same image preprocessing as Ozcelik and VanRullen [4]. Previous state-of-the-art Ozcelik and VanRullen [4] results are directly comparable to MindEye as the same test set and Versatile Diffusion model were used. Bold indicates best performance within sections.

| Method | Param Count | Image Retrieval | Brain Retrieval |
|---|---|---|---|
| No ResBlocks | 873M | 0.880 | 0.820 |
| 2 ResBlocks + No Skip | 907M | 0.881 | 0.822 |
| 2 ResBlocks | 907M | 0.886 | **0.837** |
| 4 ResBlocks + No Skip | 940M | 0.836 | 0.767 |
| 4 ResBlocks* | 940M | **0.896** | 0.822 |
| 4 ResBlocks + Only CLS | 135M | 0.611 | 0.576 |

Table 2: Effects of varying the architecture of the MLP backbone on retrieval accuracy.

We also show a comparison with a 4-resblock model that maps to the final layer of CLIP (only the CLS classification token). This model has $7\times$ fewer parameters and does much worse than all other models. This indicates two things: (1) MindEye strongly benefits from a large parameter count MLP backbone and does not overfit even in the sample constrained settings of the NSD dataset, and (2) the fMRI voxels contain fine-grained information about images, allowing us to effectively predict all 257 CLIP image embeddings instead of just the CLS token.

**Training Strategies (Losses and Data Augmentations):** We observe that with InfoNCE, MindEye only does well on brain retrieval (Table 3). A similar trend was observed in Lin et al. [11]. We attribute this to InfoNCE being a one-sided loss that only optimizes for one retrieval objective. Simply replacing InfoNCE with CLIP loss significantly improves image retrieval. MixCo augmentation helps both unidirectional and bidirectional losses.

| Method | Image Retrieval | Brain Retrieval |
|---|---|---|
| InfoNCE | 0.237 | 0.784 |
| CLIP Loss | 0.837 | 0.791 |
| InfoNCE + MixCo | 0.303 | **0.856** |
| CLIP Loss + MixCo (BiMixCo) | 0.884 | 0.841 |
| SoftCLIP Loss | 0.837 | 0.816 |
| BiMixCo + SoftCLIP (MindEye)* | **0.896** | 0.822 |

Table 3: Effects of different losses and MixCo augmentation on MLP retrieval performance.

We also show the effect of training with our SoftCLIP loss. SoftCLIP improves over hard CLIP loss for brain retrieval but performs worse than BiMixCo. Our training regime combining SoftCLIP with BiMixCo gives the best image retrieval performance.

**Reconstruction Strategies:** To demonstrate the need for a separate diffusion prior, we train a version of MindEye where both contrastive and MSE losses are applied to the ouputs of the MLP backbone. We observe that this model does poorly in terms of retrieval metrics, indicating a tradeoff between retrieval and reconstruction objectives where it is difficult to learn a single embedding space that satisfies both objectives. Inspired by recent works in self-supervised learning [39–42], we decouple these losses using a separate MLP projector, where MSE loss is applied to the outputs of the MLP backbone and contrastive loss is applied to the outputs of the projector. This model does slightly worse in terms of reconstruction but is much better at retrieval. Finally, we train a model with a diffusion prior but no MLP projector. Contrastive loss is computed for the MLP backbone and MSE loss is computed for the diffusion prior. This model is comparable to high-level MindEye in terms of reconstruction but does worse in retrieval, giving further evidence of a tradeoff. Example reconstructions for these models are in Appendix Figure 8.

| Method | Low-Level | | | | High-Level | | Retrieval | |
|---|---|---|---|---|---|---|---|---|
| | PixCorr | SSIM | Alex(2) | Alex(5) | Incep | CLIP | Image | Brain |
| Only MLP Backbone | 0.119 | 0.346 | 73.8% | 84.1% | 81.5% | 82.6% | 0.133 | 0.631 |
| Backbone + Projector | 0.154 | 0.296 | 73.2% | 85.2% | 75.2% | 77.3% | 0.888 | 0.849 |
| Backbone + Prior | **0.206** | **0.303** | **92.1%** | **97.2%** | **94.8%** | **95.1%** | 0.934 | 0.901 |
| MindEye (only BiMixCo) | 0.195 | 0.290 | 91.1% | 96.6% | 93.7% | 94.4% | **0.974** | 0.942 |
| MindEye (0.33 BiMixCo)* | 0.198 | 0.302 | 91.6% | 96.8% | 94.6% | 95.0% | 0.972 | **0.960** |

Table 4: Effects of diffusion prior and MLP projector on reconstruction and retrieval metrics.

# 4 Related Work

In the 2000s, researchers demonstrated that visual information, such as spatial position [43], orientation [44, 45], and coarse image category [46, 47] could be decoded from fMRI signals using linear classifiers. With the introduction of generative adversarial networks [48], more sophisticated decoding became feasible as researchers mapped brain activity to the latent space of these models to reconstruct handwritten digits [49], human faces [50, 51], and natural scenes [52, 5, 53]. More recently, with the release of multimodal contrastive models like CLIP [7], diffusion models [54, 55] like Stable Diffusion [9], and new large-scale fMRI datasets [26], fMRI-to-image reconstructions have reached an unprecedented level of quality [4, 3, 28].

Lin et al. [11] reconstructed NSD images by mapping voxels to CLIP space (see also Wang et al. [56]) and fed outputs through a fine-tuned Lafite [32] GAN (MindEye reconstructions using Lafite in Appendix A.7). Differences from MindEye include using a convolutional model, no projector to separate contrastive loss from MSE loss, InfoNCE instead of CLIP loss, fine-tuning of a pretrained GAN, no diffusion prior, and mapping to both CLIP image and text space. Ozcelik and VanRullen [4] used a low- and high-level pipeline with Versatile Diffusion [6]. Differences include mapping to CLIP space via ridge regression, no contrastive learning or diffusion prior, and mapping to a VDVAE [57] for low-level reconstructions. Gu et al. [28] used a low- and high-level pipeline and extended on Ozcelik et al. [5] by reconstructing with IC-GAN [58]; they did not flatten voxels and mapped to SwAV [36] features with surface-based convolutional networks. Takagi and Nishimoto [3] used ridge regression to map to Stable Diffusion latents and CLIP text latents, using different voxel selections for different components. Mind-Vis [59] did not use the Natural Scenes Dataset but tackled the fMRI-to-image problem from the unique angle of first pre-training a masked brain model on a separate large-scale dataset. This enabled the authors to use a more informative latent as the input to their image reconstruction model. Mind-Video [60] subsequently extended on the Mind-Vis approach by reconstructing video rather than images. Overall, MindEye is unique in its use of reconstruction and retrieval submodules, a deep MLP backbone with 940 million parameters (parameter size comparison in Table 10), a diffusion prior for more accurate translation across brain and image modalities, and state-of-the-art reconstuction and retrieval results.

# 5   Conclusion

We present MindEye, a novel mental decoding approach that achieves state-of-the-art reconstructions of natural scenes presented to humans in the MRI machine. These reconstructions retain semantic and perceptual similarity to the original images due to the use of a combined high-level and low-level pipeline. The novel use of specialized submodules for contrastive-based retrieval and diffusion-based reconstruction allows MindEye to learn mappings for both tasks in parallel. MindEye brain latents contain fine-grained image-specific signal as demonstrated by the ability to select ground truth images out of a set of nearly 1,000 possible images (see Figure 3). We leveraged pretrained CLIP [7], a model trained with billions of image and text data samples, as a teacher to guide MindEye where we have a relative scarcity of brain data (less than 30,000 training samples per participant). Our diffusion prior submodule is trained from scratch and allows for accurate translation of brain embeddings into pretrained CLIP space such that any model that accepts CLIP image embeddings can be provided with CLIP fMRI embeddings without fine-tuning. This flexibility suggests that MindEye reconstructions will continue to improve as newer, more powerful image generation models are released.

**Privacy Concerns & Societal Benefits:** The ability to accurately reconstruct perception from brain activity prompts questions about broader societal impacts. For instance, it should be possible to generalize current reconstruction models from perception to mental imagery without training a new model [61–64]. However, current models are not capable of across-subject decoding and each NSD participant spent up to 40 hours in the MRI machine to procure sufficient training data (see Appendix 9 for results using a subset of the total training data). Furthermore, non-invasive neuroimaging methods in general require compliance because participants can easily resist decoding by moving their head or thinking about unrelated information [65]. MindEye is also limited to natural scenes such as those in MS-COCO; for other image distributions additional data collection and specialized generative models would be required. While high-quality image reconstruction via non-invasive neuroimaging is not currently practical for real-world applications, technology is constantly improving and it is important that brain data be carefully protected and companies collecting such data be transparent with their use.

Image reconstruction from brain activity can enable various potential societal benefits. Reconstructions are expected to be systematically distorted due to mental state, neurological conditions, etc. This could potentially enable novel clinical diagnosis and assessment approaches. For example, patients suffering from major depressive disorder might produce reconstructions where emotionally negative aspects of images are more salient [66]. MindEye results also suggest potential for improved locked-in (pseudocoma) patient communication via fine-grained visual communication beyond simple classification [67], as well as brain-computer interface performance if adapted to real-time fMRI analysis [68] or non-fMRI neuroimaging modalities.

**Future Directions:** Future directions we wish to explore include mapping individual subjects to a shared embedding space to enable training models that are generalizeable across people (e.g., [69, 70]), and exploring model intepretability approaches like GradCAM [71] or Network Dissection [72] to identify the fMRI voxels that most strongly respond to the presence of certain image features.

# 6   Open Research: 100% Transparent Volunteer-Driven Science

MindEye was openly developed through volunteer contributions in the MedARC Discord server. Source code was always accessible via a public GitHub repository throughout the lifespan of the project. Research discussions were held via public Discord channels, and weekly video conference calls were recorded and shared publicly. We continue to extend a global invitation to contribute to MedARC Neuroimaging & AI Lab projects to cultivate an internationally diversified, volunteer-driven research team composed of members from varied backgrounds possessing a wide array of expertise. We contend that fully transparent open-research initiatives such as this and others like EleutherAI, LAION, OpenBioML, and ML Collective could redefine the traditional framework of scientific research, democratizing entry into machine learning and medical research through the harnessing of crowd-sourced collective intelligence and community collaboration.

# 7   Author Contributions

For detailed author contributions see Appendix A.1.

# 8 Acknowledgements

Thanks to the MedARC community, including Jeremy Howard, Tommaso Furlanello, Mihir Tripathy, and Cesar Torrico for useful discussion and reviewing the manuscript. Thank you to Furkan Ozcelik, author of Brain-Diffuser, for sharing his code and expert knowledge with our group. We thank LAION for being the initial community where this project developed, and thank Romain Beaumont and Zion English for useful discussion during that time. We thank Stability AI for sharing their high-performance computing workplace and giving us the computational resources necessary to develop MindEye. Thank you to Richard Vencu for help navigating the Stability HPC. Collection of the Natural Scenes Dataset was supported by NSF IIS-1822683 and NSF IIS-1822929.

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

# A  Appendix

## A.1  Author Contributions

PSS devised the project, led the team, developed the models, drafted the manuscript, and otherwise contributed to all parts of MindEye development. AB drafted the manuscript, developed the models, and contributed to all parts of MindEye development including creating the low-level pipeline, conception of BiMixCo and soft CLIP loss, and modification of the DALL-E 2 diffusion prior. JG developed the models, tracked/compared model variants, and significantly contributed to the MindEye codebase. SS conceived of and implemented LAION-5B retrieval using the CLIP Retrieval client and conducted various exploratory experiments. AN implemented the Lafite pipeline for MindEye reconstructions. EC conducted various initial explorations into using a diffusion prior for aligning voxels to CLIP space. AJD created the initial webdatasets used to train MindEye and created various model architectures to compare different mapping approaches. NV conducted various exploratory experiments mapping voxels to StyleGAN-XL [73] latent space. EY shared code to automatically identify identical images for qualitative comparisons and added code to ensure LAION-5B retrieval did not retrieve ground truth images. DW conducted various exploratory experiments and helped with project discussions. KAN oversaw the project and contributed valuable feedback. TMA oversaw the project, conducted initial explorations using VQGAN [74], and helped keep the project on-track through MedARC and Stability AI communication.

## A.2  Additional Dataset Information

The Natural Scenes Dataset (NSD) [26] is a public 7-Tesla fMRI dataset containing the brain responses of several human participants each spending up to 40 hours in the MRI machine passively viewing images. These square-cropped images of natural scenes were sourced from the MS-COCO dataset [27]. Each of 9,000-10,000 unique images was presented for three seconds at a time, shown three times across 30-40 scanning sessions, totaling 22,000-30,000 trials of fMRI responses per participant. fMRI responses correspond to session-wise z-scored single-trial betas output from GLMSingle [75]. Following the procedure used in other reconstruction studies that used NSD [5, 28, 3], we train individual-subject models for the four participants who completed all scanning sessions (participants 1, 2, 5, and 7) and used a test set corresponding to the shared 1,000 images presented to every participant. This yields a dataset consisting of 24,980 training samples and 2,770 test samples—we average across the three same-image repetitions for the test set (leaving 982 test samples) but not the training set, similar to Takagi and Nishimoto [3]. We use preprocessed flattened fMRI voxels in 1.8-mm native volume space corresponding to the "nsdgeneral" brain region, defined by the NSD authors as the subset of voxels in posterior cortex most responsive to the visual stimuli presented (between 13,000 to 16,000 voxels per participant). MindEye was developed using a training and validation set of Subject 1's data, with the test set (and other subjects' data) untouched until final training of models.

## A.3  MindEye Architecture

PyTorch code for the MLP backbone and projector is depicted in Algorithm 1. Specifics on how we modified the open-source implementation of the DALL-E 2 diffusion prior are discussed in A.3.1.

### A.3.1  Modifications from DALL-E 2 Diffusion Prior

The inputs for the diffusion prior are 257 backbone embeddings, 1 timestep embedding, and 257 noised CLIP image embeddings, and the output is 257 denoised CLIP image embeddings. Unlike the DALL-E 2 prior, we do not use learnable queries and instead directly predict denoised CLIP embeddings from the noised embeddings. This significantly saves on memory and allows us to train the backbone and prior end-to-end on a single GPU. We observe that adding absolute positional embeddings to the noised CLIP embeddings improves performance in the absence of learnable queries. We also observe that our prior can work with just 100 timesteps instead of 1000 as used in DALL-E 2. This makes our prior much faster at inference time. We conducted experiments with both causal and bidirectional attention and did not observe any significant difference in reconstruction performance. For simplicity we use bidirectional attention in our final model.

**Algorithm 1** PyTorch code for MindEye MLP backbone and MLP projector

```python
class BrainMLP(nn.Module):
    def __init__(self, out_dim=257*768, in_dim=15724, clip_size=768, h=4096):
        super().__init__()
        # in_dim corresponds to the subject-specific
        # number of voxels in the "nsdgeneral" brain region.
        self.lin0 = nn.Sequential(
            nn.Linear(in_dim, h, bias=False),
            nn.LayerNorm(h),
            nn.GELU(inplace=True),
            nn.Dropout(0.5))
        self.mlp = nn.ModuleList([
            nn.Sequential(
                nn.Linear(h, h, bias=False),
                nn.LayerNorm(h),
                nn.GELU(inplace=True),
                nn.Dropout(0.15)
            ) for _ in range(4)])
        self.lin1 = nn.Linear(h, out_dim, bias=True)
        self.proj = nn.Sequential(
            nn.LayerNorm(clip_size),
            nn.GELU(inplace=True),
            nn.Linear(clip_size, 2048, bias=False),
            nn.LayerNorm(2048),
            nn.GELU(inplace=True),
            nn.Linear(clip_size, 2048, bias=False),
            nn.LayerNorm(2048),
            nn.GELU(inplace=True),
            nn.Linear(2048, clip_size, bias=True))
        self.clip_size = clip_size

    def forward(self, x):
        x = self.lin0(x)
        residual = x
        for res_block in range(len(self.mlp)):
            x = self.mlp[res_block](x)
            x += residual
            residual = x
        diffusion_prior_input = self.lin1(x).reshape(len(x), -1, self.clip_size)
        disjointed_clip_fmri = self.proj(diffusion_prior_input)

        return diffusion_prior_input, disjointed_clip_fmri
```

### A.3.2 Low-Level Pipeline: Mapping to Stable Diffusion Variational Autoencoder

To map to Stable Diffusion's VAE latent space we use a low-level pipeline with the same architecture as the high level pipeline. We use a separate residual MLP backbone with 4 residual blocks that maps flattened voxels to a $16 \times 16 \times 64$ dimensional latent space. The reconstruction submodule in the low-level pipeline is a CNN upsampler that upsamples these latents by $4\times$ to create embeddings of size $(64, 64, 4)$. The CNN upsampler uses a similar architecture to Stable Diffusion's VAE decoder, which does an $8\times$ upsampling. To create the targets for the upsampler we upsample NSD images to $512 \times 512$ through bilinear interpolation and encode them with the SD VAE encoder. The resulting $(64, 64, 4)$ embeddings form the targets for the high-level pipeline.

Recent works in low-level vision (super-resolution, denoising, deblurring, etc.) have observed that mean absolute error performs better than mean squared error for pixel-level metrics like PSNR and SSIM [76, 77] due to better convergence properties. It has been shown that the 4-channel SD latent space effectively compresses images, and latents can be converted to RGB images with a linear mapping from latent space to pixel space [78]. We observe that the problem of mapping to SD embedding space follows the same properties as low-level vision tasks, such that mean absolute error performs better than mean squared error. We also experiment with using a "full reconstruction" loss where we reconstruct complete images using the SD VAE decoder and apply the loss in pixel space. This performs worse than only applying the loss in latent space and also requires significantly more GPU memory.

The contrastive submodule in the low-level pipeline acts as an auxiliary loss to improve the performance of the reconstruction submodule. It uses an MLP projector that maps the $(16, 16, 64)$ backbone outputs to $(16, 16, 512)$. Since we do not care about retrieval performance for the low-level pipeline, we simply use SoftCLIP loss without BiMixCo. To maximize low-level performance we distill the knowledge of VICRegL [79] ConvNext-XXL instead of CLIP ViT. VICRegL with $\alpha = 0.75$ is specialized for low-level tasks and achieves state-of-the-art linear segmentation results, unlike CLIP which has been trained with high-level text guidance.

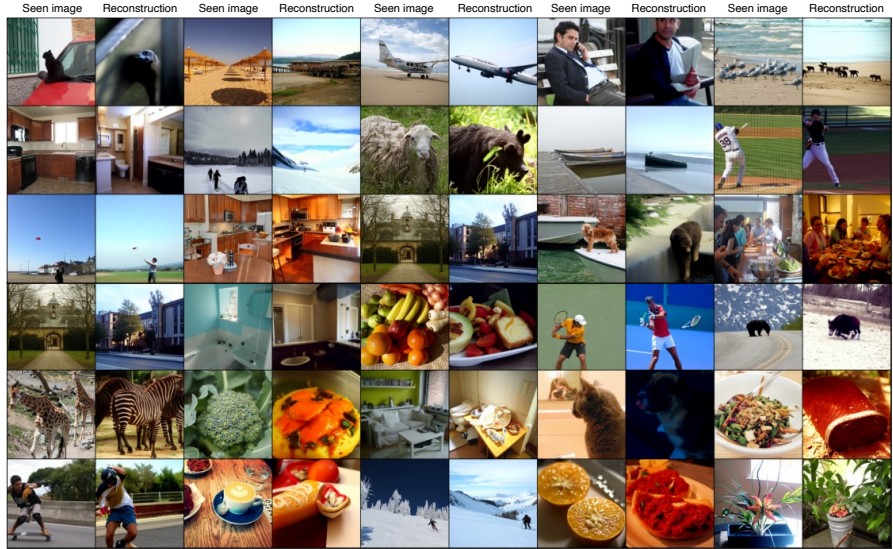

Figure 5: Additional MindEye reconstructions for Subject 1, randomly selected.

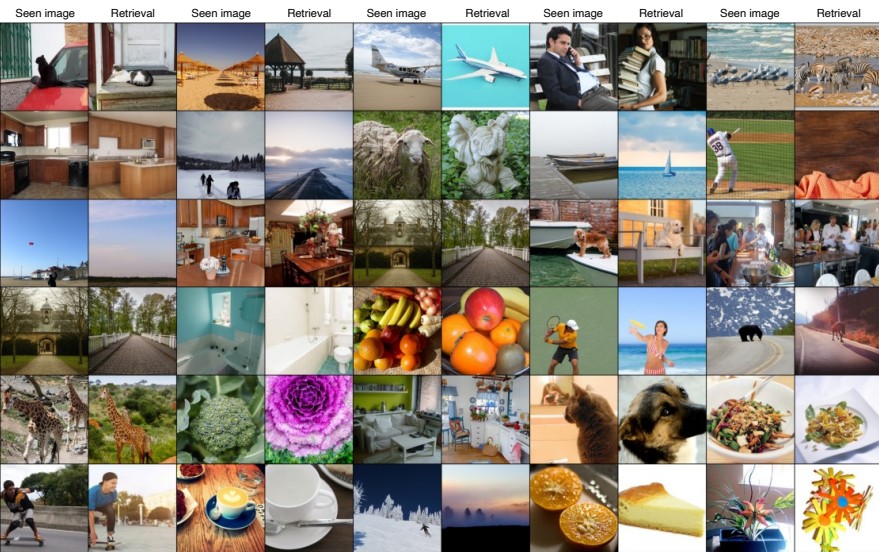

Figure 6: Additional MindEye retrievals from LAION-5B for Subject 1, randomly selected.

## A.4    More Reconstructions / Retrievals

Images containing all 982 reconstructions and retrievals for each subject are on GitHub. Figure 5 depicts a subset of randomly selected reconstruction examples from Subject 1 (first try random selection of 30 samples). Figure 6 likewise depicts randomly selected examples from LAION-5B retrieval. Figure 7 depicts randomly selected example reconstructions from the low-level pipeline. Figure 8 depicts randomly selected reconstructions for the models described in 3.3.

## A.5    UMAP Comparison

As depicted in Figure 9, the CLIP image and CLIP fMRI embedding spaces are disjointed before being fed through the diffusion prior. While the MLP projector does improve alignment compared to the outputs of the MLP backbone, the diffusion prior does a much better job at aligning the two spaces as shown by decreased euclidean distance between data points following UMAP dimensionality reduction.

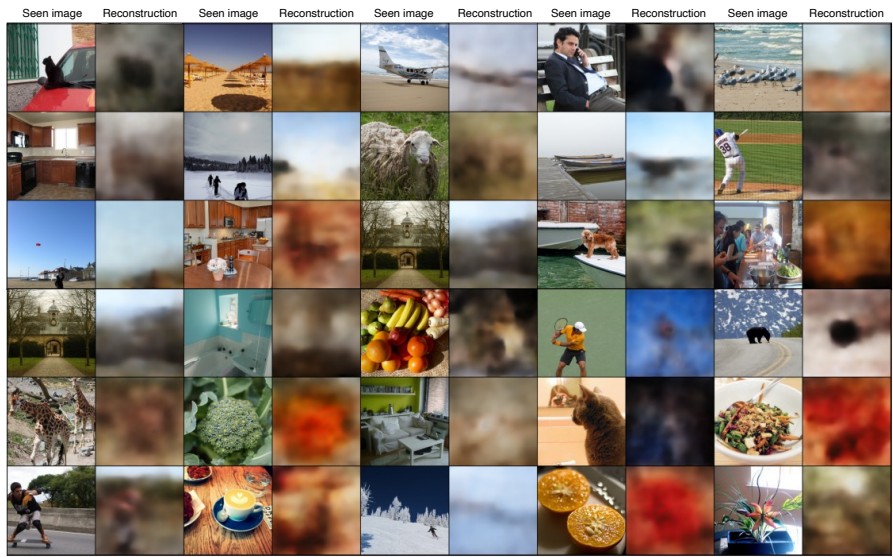

Figure 7: Example MindEye reconstructions for Subject 1 output from the low-level pipeline.

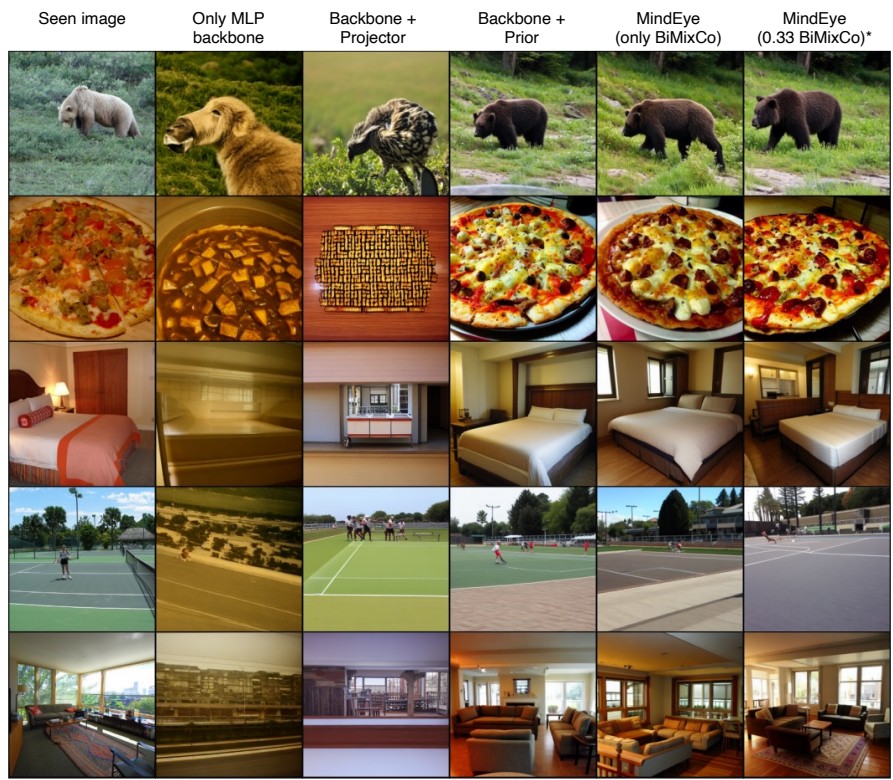

Figure 8: Example reconstructions for ablation models from Table 4

| Img2Img Strength | Low-Level | | | | High-Level | | | |
|---|---|---|---|---|---|---|---|---|
| | PixCorr ↑ | SSIM ↑ | Alex(2) ↑ | Alex(5) ↑ | Incep ↑ | CLIP ↑ | Eff ↓ | SwAV ↓ |
| 1.0 (Only low-level) | **.456** | **.493** | 87.1% | 84.1% | 61.6% | 62.4% | .992 | .638 |
| 0.7 | .439 | .416 | 92.7% | 95.1% | 90.0% | 87.5% | .803 | .514 |
| 0.5 | .429 | .389 | 96.3% | 98.4% | **94.7%** | 92.3% | .674 | .405 |
| 0.3 | .410 | .358 | **97.5%** | **98.8%** | 94.7% | 94.5% | .638 | .362 |
| 0.15* | .390 | .337 | 97.4% | 98.7% | 94.5% | 94.6% | **.630** | **.358** |
| 0.0 (Only high-level) | .209 | .318 | 92.8% | 98.0% | 94.5% | **94.8%** | .635 | .361 |

Table 5: Evaluations from Subject 1 varying img2img strength from 0 (no img2img) to 1 (only low-level pipeline). The final MindEye uses an img2img strength of 0.15.

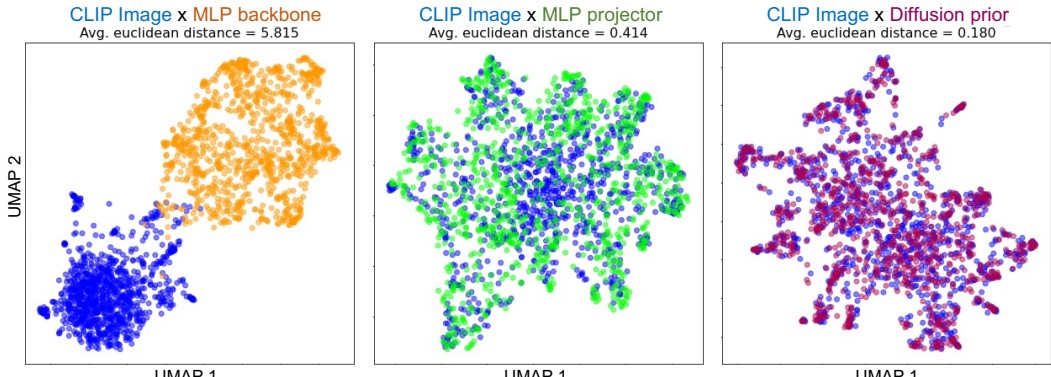

Figure 9: UMAP plots depict CLIP image latents (blue), MindEye MLP backbone latents (orange), MindEye MLP projector latents (green), and MindEye diffusion prior latents (red). UMAPs were estimated from 1,000 random samples from Subject 1. CLIP image latents correspond to the last hidden layer of ViT-L/14. Euclidean distance between the given MindEye embedding space and CLIP image space is lowest for the diffusion prior, suggesting that the diffusion prior helps to align the two embedding spaces.

## A.6 Reconstruction Evaluations: Additional Information

Two-way identification was performed in the same manner as Ozcelik and VanRullen [4]. For each model, we computed the Pearson correlation between embeddings for the ground truth image and the reconstructed image, as well as the correlation between the ground truth image and a different reconstruction elsewhere in the test set. If the correlation for the former was higher than the latter, this was marked as correct. For each test sample, performance was averaged across all possible pairwise comparisons using the other 981 reconstructions to ensure no bias from random sample selection. This yielded 982 averaged percent correct outputs, which we averaged across to obtain the metrics reported in Table 1.

Retrieval evaluations for Ozcelik and VanRullen [4] were not reported in the original paper; we calculated image/brain retrieval ourselves with the help of the Brain-Diffuser GitHub repository.

## A.7 Reconstructions from Stable Diffusion (Image Variations) and Lafite

We also attempted reconstructions using Stable Diffusion (Image Variations) [31] and Lafite [32] rather than Versatile Diffusion. Reconstructions from these models for Subject 1 are depicted in Figure 10, with metrics reported in Table 6.

For Stable Diffusion (Image Variations) we use the same approach as MindEye + Versatile Diffusion except we map from voxels to the $1 \times 768$ final layer outputs of ViT-L/14 (same architecture as "4 ResBlocks + Only CLS" in Table 2). For the diffusion prior we fine-tune an open-sourced implementation of the DALL-E 2 prior that was trained to generate CLIP image embeddings from CLIP text embeddings using 250M image-caption pairs from LAION-Aesthetics. We note that using this pretrained prior works much better than training from scratch, suggesting that a similar large-scale pretrained prior for Versatile Diffusion might further improve fMRI reconstructions. We

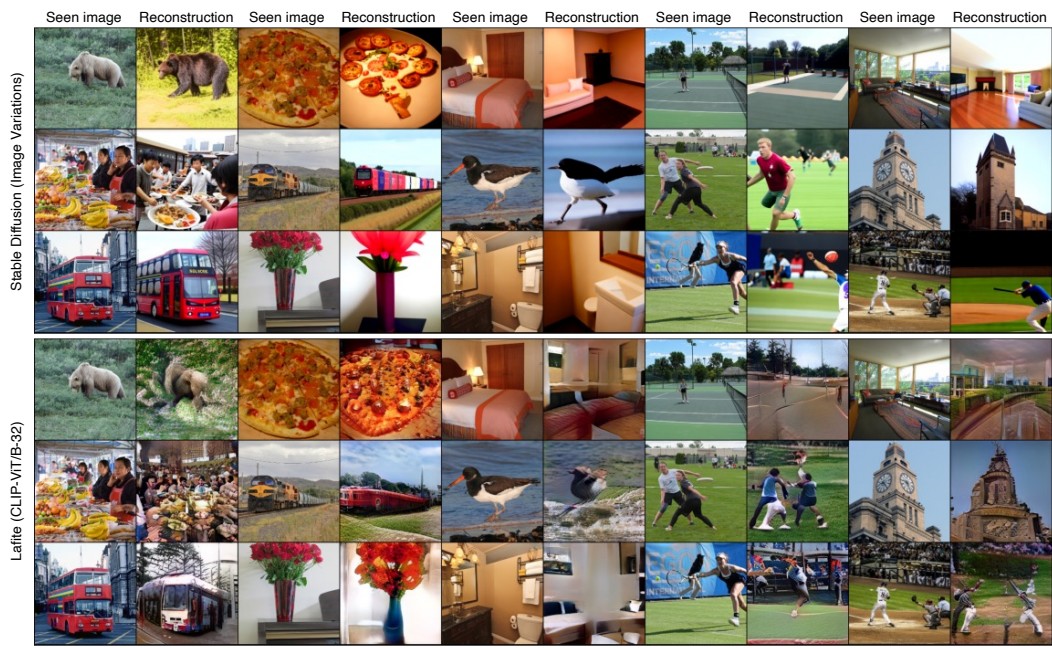

Figure 10: Corresponding reconstructions to Figure 1 when swapping Versatile Diffusion with Stable Diffusion (Image Variations) or Lafite.

use this MindEye model variant both for reconstructing via Stable Diffusion (Image Variations) and for retrieving the top-16 nearest neighbors in CLIP space for LAION-5B image retrieval. This is because the CLIP Retrieval client [30] only has precomputed CLIP embeddings for the final layer of CLIP, not the last hidden layer as used by Versatile Diffusion.

For Lafite we tried to replicate the same approach as Lin et al. [11] but with inputs from the MindEye MLP backbone. Lafite is a conditional image generation pipeline that uses the CLIP-aligned voxel embeddings as "condition vectors". In particular, Lafite leverages a StyleGAN that uses the CLIP embeddings as "style" vectors to generate images. Lafite's discriminator is trained to distinguish generated images from ground truth images and also to semantically align the CLIP embedding of the generated image with the condition vector using contrastive learning. Here we train two mapping models $f_{mi}$ and $f_{mc}$ that map voxels to the final layer of CLIP ViT-B/32, where $f_{mi}$ is contrastively aligned with CLIP image embeddings and $f_{mc}$ is contrastively aligned with CLIP text embeddings. We used the same contrastive learning schedule as MindEye with BiMixCo for the first one-third of the training cycle and SoftCLIP for the rest. Note that Lafite doesn't require training a prior so we only train the MLP backbone. Once the mapping models $f_{mi}$ and $f_{mc}$ are trained, we follow Lin et al. [11] to fine-tune a pretrained language-free Lafite model provided by [32]. Finally, we use a low-level "perceptual" pipeline by aligning layer-2 ResNet features of the generated image with those of the ground truth image using contrastive learning. The ResNet was trained using a self-supervised VICReg loss [80].

| Method | Low-Level | | | | High-Level | | | |
|---|---|---|---|---|---|---|---|---|
| | PixCorr ↑ | SSIM ↑ | Alex(2) ↑ | Alex(5) ↑ | Incep ↑ | CLIP ↑ | Eff ↓ | SwAV ↓ |
| Versatile Diffusion (S1) | **.390** | .337 | **97.4**% | **98.7**% | **94.5**% | **94.6**% | **.630** | **.358** |
| SD Image Variations (S1) | .376 | **.350** | 95.7% | 96.4% | 92.5% | 92.5% | .734 | .446 |
| Lafite (S1) | .241 | .304 | 92.5% | 98.1% | 93.7% | 87.0% | .701 | .436 |

Table 6: Evaluations for Subject 1 across three pretrained final image generation models (Lafite was fine-tuned in the same manner as Lin et al. [11]). Both Versatile Diffusion and Stable Diffusion (image variations) used an img2img strength of .15 with the low-level reconstructions output from MindEye (Lafite is a GAN and not compatible with the same img2img process).

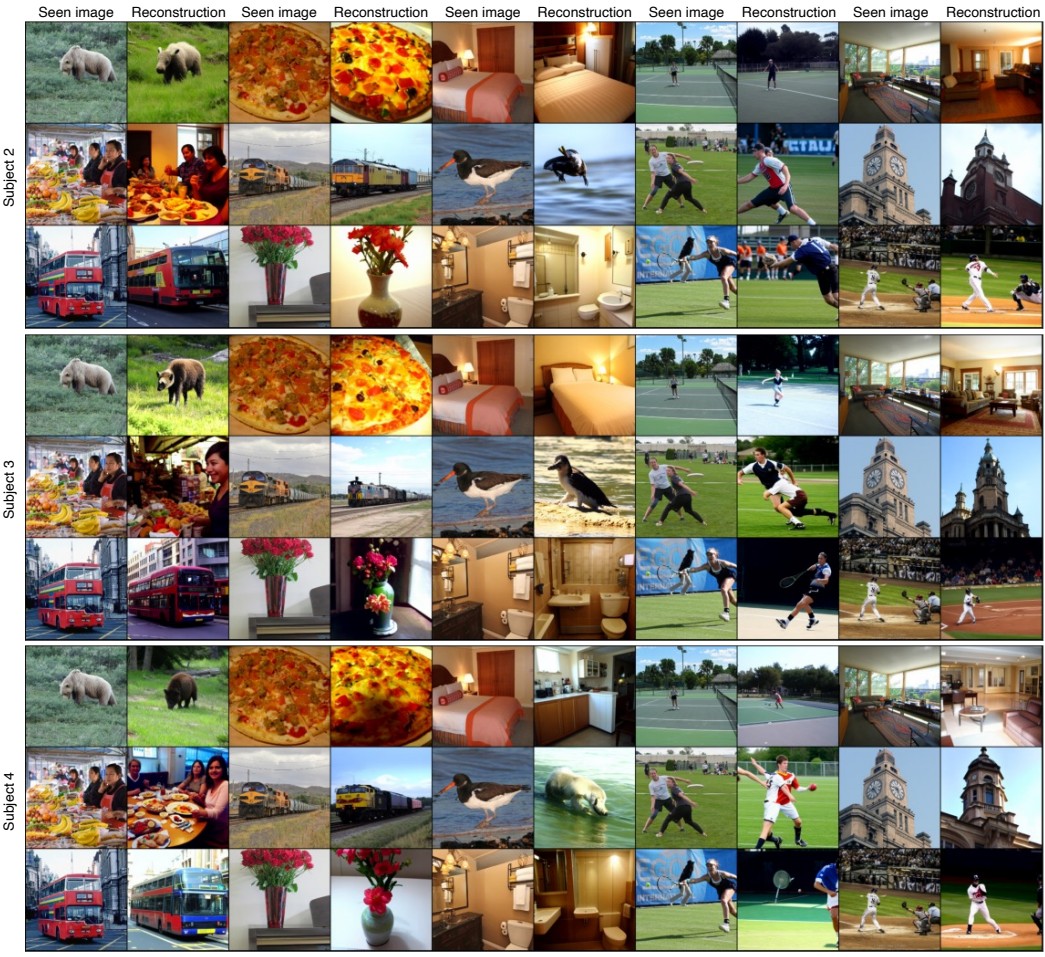

Figure 11: Corresponding reconstructions to Figure 1 for the other 3 NSD participants.

## A.8 Subject-Specific Results

Here we depict reconstructions across the other 3 NSD participants in Figure 11, with individual subject evaluations metrics in Table 7.

| Method | Low-Level | | | | High-Level | | | | Retrieval | |
|---|---|---|---|---|---|---|---|---|---|---|
| | PixCorr ↑ | SSIM ↑ | Alex(2) ↑ | Alex(5) ↑ | Incep ↑ | CLIP ↑ | Eff ↓ | SwAV ↓ | Image ↑ | Brain ↑ |
| MindEye (Subj 1) | .390 | .337 | 97.4% | 98.7% | 94.5% | 94.6% | .630 | .358 | 97.2% | 94.7% |
| MindEye (Subj 2) | .318 | .327 | 95.8% | 98.1% | 93.2% | 93.7% | .656 | .368 | 97.1% | 93.9% |
| MindEye (Subj 3) | .265 | .311 | 93.2% | 97.8% | 94.9% | 94.9% | .628 | .353 | 90.7% | 85.7% |
| MindEye (Subj 4) | .261 | .316 | 92.3% | 96.6% | 92.4% | 93.0% | .666 | .387 | 89.4% | 85.9% |

Table 7: MindEye retrieval and reconstruction performance for individual participants. These scores were averaged across participants for the values shown in Table 1.

## A.9 Single-Trial Results

In the main paper we report results from the test dataset following the standard approach of averaging voxels across the three same-image repetitions. Reconstruction evaluations using only one brain sample for each image is shown in Table 8, with example reconstructions in Figure 12.

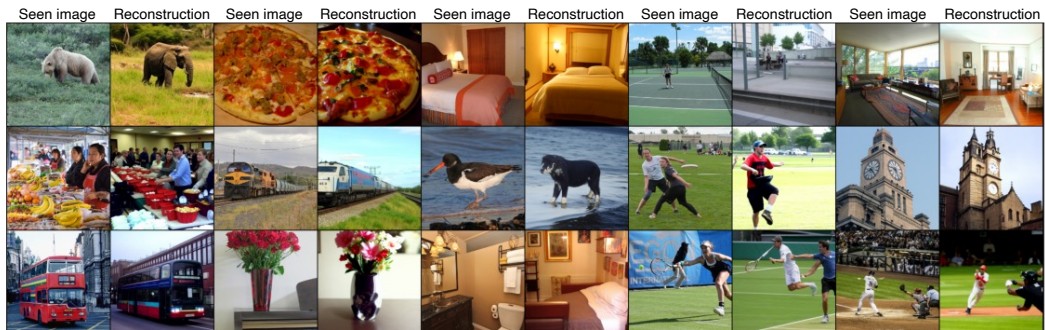

Figure 12: Corresponding reconstructions to Figure 1 using brain activity from only the first sample of every image. This is in contrast to Figure 1 which reconstructed from brain activity averaged across three same-image repetitions.

| Method | Low-Level | | | | High-Level | | | | Retrieval | |
|---|---|---|---|---|---|---|---|---|---|---|
| | PixCorr ↑ | SSIM ↑ | Alex(2) ↑ | Alex(5) ↑ | Incep ↑ | CLIP ↑ | Eff ↓ | SwAV ↓ | Image ↑ | Brain ↑ |
| MindEye | .309 | .323 | 94.7% | 97.8% | 93.8% | 94.1% | .645 | .367 | 93.6% | 90.1% |
| MindEye (single-trial) | .255 | .308 | 91.6% | 95.9% | 91.3% | 91.6% | .691 | .398 | 80.3% | 77.6% |
| MindEye (single-, S1) | .329 | .323 | 94.8% | 97.3% | 92.8% | 92.7% | .680 | .387 | 89.0% | 86.5% |
| MindEye (single-, S2) | .267 | .311 | 93.1% | 96.9% | 91.5% | 91.2% | .687 | .398 | 88.5% | 86.1% |
| MindEye (single-, S3) | .217 | .297 | 90.2% | 96.3% | 93.2% | 94.0% | .671 | .381 | 75.3% | 71.8% |
| MindEye (single-, S4) | .209 | .302 | 88.3% | 93.1% | 87.6% | 88.7% | .727 | .427 | 68.5% | 66.1% |

Table 8: MindEye retrieval and reconstruction performance for single-trial brain activations, chosen randomly out of three possible samples per unique image. Other than using single-trial brain activity, the same settings were used as in Table 1.

## A.10 Performance with varying dataset size

| Method | Low-Level | | | | High-Level | | | | Retrieval | |
|---|---|---|---|---|---|---|---|---|---|---|
| | PixCorr ↑ | SSIM ↑ | Alex(2) ↑ | Alex(5) ↑ | Incep ↑ | CLIP ↑ | Eff ↓ | SwAV ↓ | Image ↑ | Brain ↑ |
| All Data (High-Level) | .209 | .318 | 92.8% | 98.0% | 94.5% | 94.8% | .635 | .361 | 97.2% | 94.7% |
| Half Data (High-Level) | .149 | .276 | 87.7% | 94.3% | 87.1% | 90.1% | .738 | .424 | 77.5% | 60.8% |
| 2-Sessions (High-Level) | .119 | .281 | 81.0% | 88.2% | 79.2% | 84.4% | .824 | .472 | 17.9% | 12.0% |

Table 9: Quantitative comparison of MindEye performance with varying dataset sizes on Subject 1 with the high-level pipeline. Half Data corresponds to MindEye trained with half of the training samples randomly removed. 2-Sessions corresponds to MindEye trained with a random selection of 500 training image samples (or 1,500 training fMRI samples given 3 repetitions per image), equivalent to the number of samples collected across two scan sessions. Notably, image and brain retrieval metrics maintained state-of-the-art performance even when training the model with half of the training samples removed, and reconstruction performance remained competitive with previous models even with reduced training data. This suggests that our MindEye approach is flexible to being trained with smaller datasets.

## A.11 Model size comparison with other methods

| Method | | Parameter Count |
|---|---|---|
| Lin et al. | | $2 \times 1.17$M deep models + StyleGAN |
| Takagi et al. | Low Level | 37M linear regression model |
| | High Level | 450M linear regression model |
| Ozcelik et al. | Low Level | 1.45B linear regression model |
| | High Level | 257 separate 12M linear regression models |
| MindEye | Low Level | 206M residual MLP + CNN decoder model |
| | High Level | 996M residual MLP + diffusion prior model |

Table 10: Comparison of MindEye parameter count with other competing methods. Other methods primarily rely on linear regression or relatively small deep models.

