# OpenReview forum: "Reconstructing the Mind's Eye: fMRI-to-Image with Contrastive Learning and Diffusion Priors"
_NeurIPS.cc/2023/Conference — NeurIPS 2023 spotlight_

### Official Review · Reviewer_pqrP · 2023-07-05

**Soundness:** 3 good
**Presentation:** 2 fair
**Contribution:** 3 good
**Rating:** 6
**Confidence:** 4

**Summary:**

This paper presents a novel fMRI-to-image approach (MindEye) that can achieve excellent reconstructions of natural scenes. The main idea is to retrieve and reconstruct viewed images from fMRI with brain activity information. The model consists of two parallel submodules. The retrieval submodule uses contrastive loss to retrieve images and generate features with high cosine similarity with the corresponding image embeddings. In the reconstruction submodule, the diffusion prior and CLIP image embedding are used to generate aligned semantic features. In order to better reconstruct the image, a separate encoder is trained to generate low-level features, and jointly output the reconstructed images with the previous semantic features. The experimental results also demonstrate the superior performance of the method.

**Strengths:**

Interesting idea. This paper provides a novel method to achieve fMRI-to-image task, and explores this task by introducing retrieval, diffusion prior and low-level information. Aside from the framework, the experimental results also fully illustrate its performance.

**Weaknesses:**

1. In this paper, most of the modules are existing models. The novelty of the paper requires further elaboration.
2. Lack of theoretical proof, most of them are descriptions. For example, the aligned embeddings mentioned in the paper, whether there is a theoretical proof of the rationality of the aligned embeddings for image reconstruction.
3. The network structure uses multiple large-scale models, such as Clip and Diffusion, which are relatively time-consuming, while the paper lacks an explanation of the computational cost.
4. Limited datasets. Only NSD dataset is used, and the generalization of the model is not considered.
5. The experiment is valid, only the performance of each sub-module is considered. The performance after removing a certain submodule is not clear.

**Questions:**

1. How about the comparisons with other methods in terms of computational cost and model parameters?
2. For data augmentation method, is using generative methods to generate more samples an alternative?
3. Is the diffusion prior module pre-trained? Also, I wonder what is the benefit of using a diffusion prior?

**Limitations:**

Please refer to the weakness.

---

> ### Author Rebuttal · Authors · 2023-08-09
>
> > Q1: Most of the modules are existing models. The novelty of the paper requires further elaboration.
>
> A1: MindEye relies on models trained on billions of image/text samples. NSD provides <30,000 fMRI samples per participant. We argue that part of the novelty of our approach is to use models like CLIP trained with massive datasets as a teacher to guide training of our brain models where we lack data. Note our diffusion prior is trained from scratch and that other novelties of our paper are summarized in our global response.
>
> > Q2: Lack of theoretical proof ... for example, ... theoretical proof of the rationality of the aligned embeddings for image reconstruction.
>
> A2: We now discuss existing proof for the rationality of aligned embeddings:
>
> Multimodal contrastive learning will always produce disjointed embeddings because of the “modality gap” phenomenon whereby encoding modalities into a shared space restricts the effective embedding space to a narrow cone in geometric space. Liang et al. (2022) show theoretically and empirically that contrastive learning induces a modality gap phenomenon. This modality gap phenomenon explains why our CLIP image embeddings and CLIP fMRI embeddings are isolated to different regions of the same shared embedding space (UMAP plots in Appendix A.4). We use pre-trained models that expect CLIP image embeddings as input, thus motivating our training of a diffusion prior to align disjointed embeddings.
>
> > Q3: How about the comparisons with other methods in terms of computational cost and model parameters?
>
> A3: MindEye is modular and doesn’t need all the large scale models at all times. CLIP is used only during training. Similarly, Versatile Diffusion is only needed during inference. Even though our parameter count is high, MindEye can be trained on a single A100 in less than 18 hours. This is because our models primarily consist of MLPs which are faster to compute than transformers or CNNs.
>
> At inference time, the diffusion prior can be dropped if only retrieval is needed. As stated in Appendix A.2.1, our diffusion prior is also faster than the DALLE-2 diffusion prior as it only needs 100 timesteps instead of 1000. It is also more computationally efficient because we modified the architecture to not have learnable queries and to instead directly predict denoised CLIP embeddings. For reconstruction, any off-the-shelf image generation method that accepts CLIP embeddings can be used, depending on computational cost requirements.
>
> We have added a new table to Appendix A.10 comparing MindEye’s parameter counts with other methods.
>
> > Q4: Limited datasets. Only NSD dataset is used, and the generalization of the model is not considered.
>
> A4: We agree that generalization is important – our ultimate goal is to show results that are comparable to our NSD findings in other, independently-collected datasets. The main challenge here is that all other fMRI datasets of this kind are much smaller than NSD, by an order of magnitude, and the sheer size of NSD is a major contributor to the very high quality of the results shown here (see Appendix A.9, attached as a pdf to our global response, to see how performance scales as a function of training set size). To obtain comparable performance on smaller datasets, we will need to find a suitable way of combining data across subjects; this would allow us, e.g., to train on large datasets like NSD and then fine-tune and test on a smaller dataset incorporating different subjects.  This “across-subject alignment” problem in fMRI data analysis is difficult because subjects differ in brain structure (which can lead to different input dimensionalities and potential misalignment across voxels) and in life experiences (which can lead to functional organization of visual concepts in their brains being different, even if their brains are structurally aligned). For the present work, we focused on optimizing single-subject decoding in the “high data” setting of NSD; going forward, we will be changing the model architecture to explicitly learn a shared-subject embedding space that supports across-subject decoding (we now discuss this in the Conclusions). Based on other alignment findings in the fMRI literature (e.g., “shared response modeling”, Chen et al., 2015), we think this approach has extensive promise. However, implementing the shared embedding space in MindEye requires substantially more work to complete and we do not have results yet for this exploration.
>
> > Q5: The performance after removing a certain submodule is not clear.
>
> A5: Performance with and without each of the 2 submodules is mentioned in Table 4. We also show the corresponding reconstructions for these ablations in Figure 8 in the Appendix. We also show retrieval and reconstruction performance when using just the backbone, backbone + projector, backbone + prior, and backbone + prior + projector.
>
> > Q6: For data augmentation method, is using generative methods to generate more samples an alternative?
>
> A6: Our model needs paired fMRI and visual stimulus data for which there are limited datasets. Establishing generative models for fMRI data is an important research problem in its own right, and at present there are not well-established methods for doing this. This is why we use mixup to create synthetic fMRI responses for MindEye.
>
> We considered augmenting image samples by randomly replacing the prior’s target with embeddings of generated image variations, instead of the ground truth image. We expect this could make the prior better at modeling the distribution of target images. But this augmentation would also slow down the training pipeline considerably.
>
> > Q7: Is the diffusion prior module pre-trained? Also, I wonder what is the benefit of using a diffusion prior?
>
> A7: The diffusion prior is not pre-trained. We show benefits of using a diffusion prior in Table 4. Figure 9 in the Appendix shows UMAP projections of CLIP image embeddings before and after the diffusion prior.

---

> > ### Comment · Reviewer_pqrP · 2023-08-21
> >
> > I appreciate the authors' rebuttal which clarifies some of my concerns. While the methodology can be further improved, the paper has its scientific merits. After reading the detailed responses, I tend to raise my score to weak accept.

---

### Official Review · Reviewer_Pn57 · 2023-07-06

**Soundness:** 3 good
**Presentation:** 4 excellent
**Contribution:** 3 good
**Rating:** 8
**Confidence:** 4

**Summary:**

The authors present MindEye, an innovative fMRI-to-image approach that utilizes contrastive learning and a diffusion prior for retrieval and reconstruction tasks. They conduct thorough comparisons, establishing MindEye's superiority over existing methods in terms of both qualitative and quantitative evaluations. The authors attribute the success of MindEye to its specialized submodules, improved training techniques, and models with increased parameterization. Additionally, they showcase MindEye's ability to preserve low-level image features in reconstructions through the use of img2img. Overall, MindEye represents a significant advancement in the field, pushing the boundaries of fMRI-based image retrieval and reconstruction.

**Strengths:**

The authors conducted a comprehensive comparison of MindEye with existing methods, employing both qualitative side-by-side comparisons and quantitative evaluations. The results demonstrate that MindEye achieves state-of-the-art performance in both reconstruction and retrieval tasks. Notably, MindEye excels at retrieving the exact original image even when faced with highly similar candidates, indicating that its brain embeddings retain fine-grained, image-specific information. This remarkable capability enables accurate image retrieval even from extensive databases such as LAION-5B. The paper is well written and the presentation is to the point.

**Weaknesses:**

The methodology employed in this study has some limitations in terms of originality, as a majority of the approach relies on external state-of-the-art models. The authors primarily train simple MLPs and utilize a pre-trained diffusion prior.

**Questions:**

1. Exploring the applicability of the model to different patients would yield valuable insights. It raises the question of whether the model can provide meaningful results beyond the specific patient it was trained on.

2. A sensitivity analysis investigating the relationship between image output and fMRI input would be highly intriguing. It would shed light on the crucial components of the input that contribute to generating the CLIP embedding and ultimately influence the quality of the reconstructed image.

3. The authors suggest that increasing the number of parameters improves the results. It would be informative to know if they experimented with deeper networks to explore the potential benefits of a more complex architecture.


**Limitations:**

1. Lack of Methodological Originality: The study relies heavily on external state-of-the-art models, which diminishes the originality of the methodology. The authors predominantly employ simple MLPs and a pre-trained diffusion prior, which limits the novelty of their approach.

2. Applicability to Different Patients: Exploring the generalizability of the model to diverse patient populations would be valuable. It is essential to understand if the model can yield meaningful results beyond the specific patient it was initially trained on.

3. Sensitivity Analysis: Conducting a sensitivity analysis would provide valuable insights into the relationship between image output and fMRI input. Understanding which specific components of the input are crucial for generating the CLIP embedding and influencing the quality of the reconstructed image would enhance the understanding of the model's behavior.

4. Deeper Networks and Parameterization: The authors suggest that increasing the number of parameters improves the results. It would be beneficial to know if they explored the use of deeper networks, as a more complex architecture may have potential benefits. Investigating the effects of different network depths could shed light on the impact of model complexity on performance.

---

> ### Author Rebuttal · Authors · 2023-08-09
>
> Thank you for your comments and feedback!
>
> > Q1: Lack of Methodological Originality: The study relies heavily on external state-of-the-art models, which diminishes the originality of the methodology. The authors predominantly employ simple MLPs and a pre-trained diffusion prior, which limits the novelty of their approach.
>
> A1: We now clarify that our diffusion prior is trained from scratch in the “Diffusion Prior” section. That is, our diffusion prior is trained from scratch to align our disjointed CLIP fMRI embeddings, which are then fed to a pretrained Versatile Diffusion model to output image reconstructions.
>
> MindEye relies on external state-of-the-art models that were trained on billions of image and text data samples. Critically, none of these existing models were trained with brain data and NSD provides less than 30,000 training samples per participant, which is orders of magnitude fewer data points than used for models like CLIP or Versatile Diffusion. We argue that part of the novelty of our approach is to leverage models like CLIP that were trained with massive datasets as a teacher to guide the training of our brain models where we have a relative scarcity of data. We have now clarified this nuance in the “Conclusions” section.
>
> > Q2: Applicability to Different Patients: Exploring the generalizability of the model to diverse patient populations would be valuable. It is essential to understand if the model can yield meaningful results beyond the specific patient it was initially trained on.
>
> A2: A limitation that we discuss in the paper is that all MindEye models are subject-specific, meaning that they cannot be generalized to other subjects. One reason for the subject-specificity of the model is that each person has a differently shaped / sized brain and thus different numbers of voxels, resulting in different input dimensionalities and potential misalignment across voxels. Another reason is that different people have had different life experiences, so the functional organization of visual concepts in their brains could be different even if the brains are structurally aligned. A future goal our group is tackling is to change the model architecture to explicitly learn a shared-subject embedding space to potentially support across-subject decoding; based on prior “functional alignment” findings in the fMRI literature from Haxby, Ramadge, and others (e.g., “hyperalignment” and “shared response modeling” papers; Haxby et al., 2020, Chen et al., 2015) we think this approach has extensive promise. However, implementing the shared embedding space in MindEye requires substantially more work to complete and we do not have results yet for this exploration.
>
> > Q3: Sensitivity Analysis: Conducting a sensitivity analysis would provide valuable insights into the relationship between image output and fMRI input. Understanding which specific components of the input are crucial for generating the CLIP embedding and influencing the quality of the reconstructed image would enhance the understanding of the model's behavior.
>
> A3: Model interpretability methods like GradCAM (Selvaraju et al., 2016) and Network Dissection (Bau et al., 2017) could be used to determine the fMRI voxels that most strongly respond to the presence of certain image features. Similar work has already been done using the Natural Scenes Dataset in Sarch et al (2023), where they identify the most significant image features corresponding to every voxel. Their models are the reverse of MindEye, where an input image is used to predict the fMRI response for specific brain regions. Another direction could be to visualize reconstructions from synthetic fMRI inputs where voxels outside a given brain region are set to zero and voxels inside the brain region are set to an arbitrarily high value and fed into the pretrained MindEye model. Resulting reconstructions would emphasize the image qualities important to that brain region (see Lin et al., 2022; Ozcelik et al., 2023). Such sensitivity analysis could help visualize the functional specialization of the brain and the most important voxels for reconstructions. We now discuss the above prior work and this research direction in the Conclusions section.
>
> > Q4: Deeper Networks and Parameterization: The authors suggest that increasing the number of parameters improves the results. It would be beneficial to know if they explored the use of deeper networks, as a more complex architecture may have potential benefits. Investigating the effects of different network depths could shed light on the impact of model complexity on performance.
>
> A4: We observe diminishing returns from training larger models, because of the limited dataset size. The performance when going from 2 to 4 resblocks only improves when using skip connections, and even then the improvement is around 1% with a 33M increase in parameter count (Table 2). This suggests that a larger dataset would be needed for bigger models to demonstrate noticeable improvements.

---

> > ### Comment · Reviewer_Pn57 · 2023-08-15
> >
> > Thanks for the rebuttal.
> >
> > The authors' response is in line with what I anticipated. I hold the view that the stated limitations of this study, though few, still hold merit. As highlighted by the authors, these limitations could potentially be explored further in future research based on this foundation.

---

### Official Review · Reviewer_M8r4 · 2023-07-07

**Soundness:** 3 good
**Presentation:** 4 excellent
**Contribution:** 3 good
**Rating:** 7
**Confidence:** 4

**Summary:**

This paper introduces a method for fMRI-to-image conversion that facilitates realistic image reconstruction from brain recordings. The proposed method comprises two stages: semantic (high-level) and perceptual (low-level) pipelines. In the semantic stage, fMRI voxels are projected into the image space using a Multilayer Perceptron (MLP) and a diffusion prior. Notably, the paper employs the contrastive learning of the CLIP model to map fMRI data into a pre-trained CLIP model. Subsequently, the diffusion prior is used to map CLIP text embedding to image embedding. On the other hand, in the perceptual pipeline, image embeddings are mapped using an MLP projector and a CNN decoder. The method's experimental results are derived using the NSD dataset.

**Strengths:**

The method presented in the paper is clear, and the validity of the claims is compelling.

I also appreciate the experimental evaluation conducted, which yields promising results both from a quantitative and qualitative standpoint.

The paper presents a commendable ablation analysis of various loss functions and data augmentation strategies.


**Weaknesses:**

I think the main weakness of the method is the use of pre-trained models and evaluations with large datasets only. I was wondering what will happen when we train a model without having a pretraining model.

Having employed contrastive learning in the form of CLIP, with other proposed modules, I think the overall novelty of the paper might not be very high. However, I think the main contribution of the paper is clear and robust.

The presented method is quite large which makes it hard for building it with restricted resources.


**Questions:**

I wonder how the proposed model manages the domain shift when the structure of the training functional magnetic resonance imaging (fMRI) data significantly differs from that of the downstream task.

Regarding the architectural improvement ablation, what about potential implications of training the overarching concept using a more compact architecture with fewer parameters. It would be interesting to explore if a streamlined model could yield similar or improved results while offering more efficient computational processing and potentially mitigating the risk of overfitting.

What about the computational complexity of the proposed method, specifically concerning image generation? How does this model compare to other methods in terms of computational demands?

The computation complexity would be crucial for understanding its practical applicability, particularly in settings where computational resources may be limited.

Finally, I am curious about the choice of dataset. The paper utilizes NSD dataset for model training and evaluation. Could the model generalize well when applied to different datasets, specifically smaller ones? Moreover, when the dataset's size is reduced, can the model maintain its performance and possibly outperform comparative methods?


**Limitations:**

Yes, the paper discusses the possible limitation of the current work.

---

> ### Author Rebuttal · Authors · 2023-08-09
>
> > Q1: I think the main weakness of the method is the use of pre-trained models ... the overall novelty of the paper might not be very high.
>
> A1: MindEye relies on external state-of-the-art models that were trained on billions of image and text data samples. Critically, none of these existing models were trained with brain data and NSD provides less than 30,000 training fMRI samples per participant, which is orders of magnitude fewer data points than used for models like CLIP or Versatile Diffusion. Part of the novelty of our approach is to leverage models like CLIP that were trained with massive datasets as a teacher to guide the training of our brain models where we have a relative scarcity of data.
>
> > Q2: The presented method is quite large which makes it hard for building it with restricted resources. What about the computational complexity ...
>
> A2: Even though the parameter count for MindEye is high, MindEye (including both high- and low-level pipelines) can be trained on a single A100 in less than 18 hours. This efficiency is due to the bulk of the parameters stemming from MLPs, which are faster to compute than transformers or CNNs. We now mention these details in the “Diffusion Prior” section.
>
> Regarding image reconstruction, our generative model is pre-trained. Any off-the-shelf image generation method that accepts CLIP embeddings can be used with MindEye, meaning more efficient models can be swapped in depending on computational cost requirements.
>
> Also note that at inference time, our diffusion prior (which is trained from scratch) can be dropped if only retrieval is needed, and as stated in Appendix A.2.1, our diffusion prior is faster than the DALLE-2 diffusion prior because we use 100 timesteps instead of 1000, and we modified its architecture to no longer have learnable queries and to instead directly predict denoised CLIP embeddings.
>
> Regarding comparison of computational demands with other reconstruction papers, we have added a new table to Appendix A.10 (see pdf attached to global response) comparing MindEye’s parameter counts with other methods. Lin et al. (2022) use 2 CNN mapper networks of size 1.17M each and also finetune a Lafite based StyleGAN. Takagi et al. (2022) use a linear model with 450M params for their high level pipeline and a linear model with 37M params for low level. Ozcelik et al. (2023) use 257 individual linear models of 12M params each for their high level pipeline and a linear model with 1.45B params for low level. In contrast, MindEye uses an MLP + diffusion prior approach totaling 996M params for the high level pipeline and an MLP with 206M params for our low level pipeline.
>
> > Q3: Could the model generalize well when applied to different datasets, specifically smaller ones? I wonder how the proposed model manages the domain shift ...
>
> A3: As an initial exploration into the importance of dataset size to MindEye performance, we now report performance for Subject 1 using models trained on reduced subsets of the complete training data in Appendix A.9 (attached pdf to our global response).
>
> These results show that even with half the training samples, MindEye still achieves state-of-the-art retrieval performance and competitive reconstruction performance. Even with just 500 training image samples (less than 6% of the full training data), results remained competitive with previous models (albeit no longer state-of-the-art), as shown via the “2-Sessions” model evaluations.
>
> This suggests that MindEye is viable for datasets with much less data than NSD, although future work is needed to directly test performance of this approach to new datasets.
>
> Regarding the topic of domain shift, a limitation that we discuss in the paper and which will be the focus of future work is that all MindEye models are subject-specific, meaning that we cannot train a model on NSD for Subject 1 and then evaluate that model on other subjects or non-NSD datasets. One reason for this is that each person has a differently shaped / sized brain and thus different numbers of voxels, resulting in different input dimensionalities and potential misalignment across voxels. Another reason is that different people have had different life experiences, so the functional organization of visual concepts in their brains could be different even if the brains are structurally aligned. Training on NSD and evaluating on non-NSD datasets would require that the same subject was likewise scanned in the separate fMRI dataset.
>
> > Q4: What about potential implications of training the overarching concept using a more compact architecture with fewer parameters?
>
> A4: Regarding the concern of overfitting with large models, note that our current MindEye model does not overfit despite its large size thanks to our training techniques. That said, we agree it is worthwhile to explore how more compact architectures could be used to achieve similar or improved model performance with reduced parameter count.
>
> In our current approach, we predict all CLIP embeddings (size 257x768) from a hidden representation of size 4096, using a linear layer. This layer alone accounts for 808M parameters out of the 940M parameters in our backbone. Our initial attempts to reduce the size of this layer performed worse than the simple linear layer.
>
> In future works, we will explore other methods to reduce the size of this layer by compressing the information in the 257 CLIP-image tokens into a smaller number of tokens. Another possible direction could be to reduce the size of the CLIP embedding dimension using PCA (see Ramesh et al., 2020). We now mention this future direction in the Conclusions section of the paper. Note that we cannot simply use the CLS token (size 1x768) as it does not contain all the necessary information (especially low-level image information) about the image. This is demonstrated by the inferior retrieval scores from the CLS token variant of MindEye as shown in Table 2.

---

> ### Comment · Reviewer_M8r4 · 2023-08-20
>
> While I appreciate the author's thoughtful rebuttal, I maintain my original score for the following reasons:
>
> - "Each person has a differently shaped/sized brain and thus different numbers of voxels." There are several standardization methods available to unify the number of voxels. For further evaluation of this method, one possible approach could be to use a surface map, which has been employed in some studies to standardize differently shaped brains.
>
> - "Part of the novelty of our approach is to leverage models like CLIP that were trained with massive datasets as a teacher to guide the training of our brain models." While it's true that using a model trained on a massive dataset adds an element of novelty, I believe it also makes the innovation incremental and limits the model's application to scenarios where a pre-trained model is available. Conducting between-subject experiments could help us understand whether the model is capable of generalizing well across different domains.
>
> For these reasons, I stand by my initial assessment and disagree with some of the points raised in the responses.

---

### Official Review · Reviewer_DnYs · 2023-07-09

**Soundness:** 3 good
**Presentation:** 4 excellent
**Contribution:** 3 good
**Rating:** 7
**Confidence:** 5

**Summary:**

The paper improves the fMRI reconstruction method using contrastive learning strategy and diffusion prior model. The concept is relative simple but the details of the proposed method, which is the key to make difference, is well-implemented. First, the BiMixCo implements a contrastive loss between the fMRI voxel representation and the CLIP image representation with the utilization of mixup augmentation of the fMRI vectors. Second, the Diffusion prior is implemented on the fMRI representation after the MLP backbone (which is the input to the MLP projector of which output is the input the contrastive loss) for the generating the representation for reconstructing the image. Moreover, the paper implements additional low-level pipeline based on VAE and Stable Diffusion so that an initialization of reconstruction can be attained. The overall reconstruction and retrieval performance significantly outperforms the baselines with large margin, particularly for the retrieval performance.

**Strengths:**

- While the problem is not new and the proposed methods are clever combinations of existing techniques rather than brand new ones, the overall fMRI-based reconstruction and retrieval performances are impressive.
- Both retrieval and reconstruction results are presented with thorough ablation studies to justify the modeling choice.
- The visualization results are compelling and convincing.

**Weaknesses:**

- The performance gaps between MindEye and other baselines are vast while those for reconstructions are not as much. It seems like the reconstruction results for the baselines are copied from the original papers, while the retrieval results are reproduces..Is that correct? What is the main reason for such huge difference? The representations used for reconstruction and retrieval should not be that different, so such a wide gap is little mysterious. In case the retrieval results for the baselines are reproduces, why not also reproduce the results for the reconstructions?
- The effect of BiMixCo is not very convincing since as shown in Table 4, it seems to be helpful only for the retrieval task while not as much for the reconstruction task. Would there be some additional justification?


**Questions:**

Please see my comments in Weakness.

**Limitations:**

- The exact process for the evaluation for the retrieval task is not very clearly described.
- The performance gap between the reconstruction and retrieval is not clearly explained.

---

> ### Author Rebuttal · Authors · 2023-08-09
>
> > Q1: The performance gap between the reconstruction and retrieval is not clearly explained. The performance gaps between MindEye and other baselines are vast while those for reconstructions are not as much. It seems like the reconstruction results for the baselines are copied from the original papers, while the retrieval results are reproduces..Is that correct? What is the main reason for such huge difference? The representations used for reconstruction and retrieval should not be that different, so such a wide gap is little mysterious. In case the retrieval results for the baselines are reproduces, why not also reproduce the results for the reconstructions?
>
> A1: The related papers all shared the goal of reconstructing images from brain activity, not retrieving images. The one exception is Lin et al. (2022), which did report retrieval performance in their paper which is shown in our Table 1 (aka this is copied from their paper, not reproduced). We had to reproduce the retrieval results for Ozcelik et al. because the original paper reported reconstruction performance but not retrieval performance (this is mentioned in Appendix A.5).
>
> Therefore, one explanation for the discrepancy between our huge improvement to retrieval performance but only moderate improvement to reconstruction performance compared to previous work is that the authors of other papers were never aiming to improve retrieval performance. That is, other work never tried to decouple retrieval and reconstruction objectives.
>
> To get such high retrieval performance we engineered separate retrieval and reconstruction submodules. This is necessary because the objective for retrieval performance is distinct from the objective for reconstruction: minimizing cosine similarity between paired samples does not translate to minimizing mean squared error of image latents. We discuss our evidence that these objectives trade-off with each other in lines 129-130 and 239-240 of the paper (the diffusion prior’s role cannot be fulfilled by simply adding MSE loss to the MLP projector, and using both contrastive and MSE losses to the MLP backbone does not work well). In other words, representations used for reconstruction and retrieval actually are expected to be quite different.
>
> > Q2: The effect of BiMixCo is not very convincing since as shown in Table 4, it seems to be helpful only for the retrieval task while not as much for the reconstruction task. Would there be some additional justification?
>
> A2: We expect the drop in reconstruction performance due to BiMixCo to be because of how mixup generates synthetic datapoints through linear interpolation. For the retrieval task, we do not need to mix the targets as the contrastive objective optimizes the relative distance of the model predictions with positive and negative samples. However, for the reconstruction task we need absolute targets for the mixed inputs. We generated new targets by mixing the original targets in the same ratio as the mixup inputs. This causes a slight shift in the distributions of target embeddings at train time, leading to a drop in test time performance. We now clarify this in the second-to-last paragraph of the “Contrastive Learning” section. Relatedly, other recent works (Liu & Wang, 2023; Yu, Wang, & Wu, 2021) have also shown that stopping mixup after a certain number of epochs improves performance by reducing the train test disparity (this is mentioned in L107 in the paper).
>
> Also note that BiMixCo as a loss is only applied to the retrieval submodule. It doesn’t have a direct effect on the diffusion prior, except through the common MLP backbone and the slightly altered target distribution. We observe that BiMixCo gives the highest retrieval performance but slightly hurts reconstructions (Table 4). Our final schedule combining BiMixCo + SoftCLIP losses strikes the best balance between retrieval and reconstruction performance in a single model.
>
> > Q3: The exact process for the evaluation for the retrieval task is not very clearly described.
>
> A3: We have now updated the text to more clearly describe the retrieval evaluation process:
>
> “We followed the same procedure as Lin et al. [11] for calculating the retrieval metrics reported in Table 1. Brain retrieval performance was calculated according to the following procedure: for each test image, the image is converted to a CLIP image embedding and we compute the cosine similarity to both its respective ground truth disjointed CLIP fMRI embedding as well as 299 other randomly selected disjointed CLIP fMRI embeddings in the test set. For each test sample, success is determined if the cosine similarity is greatest between the ground truth CLIP embedding and its respective fMRI embedding (aka top-1 retrieval performance, chance=1/300). We average retrieval performance across all test samples and repeat the entire process 30 times to account for the variability in random sampling of batches. For image retrieval, the same procedure is used except image and brain samples are flipped such that the goal is to find the corresponding paired CLIP image embedding out of 300 possible CLIP embeddings in the batch. Lin et al. [11] refer to image retrieval as “forward retrieval” and brain retrieval as “backward retrieval” in their paper.”
>
> For context, the above paragraph will replace the current explanation on page 5 that begins with "To compare our retrieval performance to other papers we average ..."

---

> > ### Comment · Reviewer_DnYs · 2023-08-18
> >
> > Thanks for the detailed rebuttal and I will keep my original rating "Accept".

---

### Author Rebuttal · Authors · 2023-08-09

We sincerely thank the reviewers for their thorough comments, thoughts, and suggestions on our manuscript. We have done our best to answer all questions and concerns. We summarize some of the core revisions and clarifications below.

* New Appendix A.9 (see attached pdf to this response) table shows how MindEye performs given reduced subsets of the complete training data. Competitive performance was still observed even when training with less than 6% of the full training data. We also discuss how future work should explore generalization to other subjects and datasets and how this is not straightforward to immediately implement due to the nature of working with brain samples, which are intrinsically unique to the individual.
* Clarification on the novelties of our paper, especially in regards to the concern that there was a lack of novelty due to the use of large-scale pre-trained models. We note that Natural Scenes Dataset contains less than 30,000 training samples compared to the billions of samples used to train CLIP and Versatile Diffusion, and that figuring out the best way to leverage these larger models as teachers to guide the training of our brain models with relative scarcity of data is one of the novelties of this paper.
* Clarification that our diffusion prior is actually trained from scratch, and is much faster during inference than the DALLE-2 diffusion prior. This is because we only use 100 timesteps instead of 1000 timesteps, and we also implemented architectural changes (Appendix A.2.1), including no learnable queries and direct prediction of denoised CLIP embeddings, which allow the diffusion prior to be performed on a single A100. We also clarify that MindEye can work with any off-the-shelf image generation method that accepts CLIP embeddings depending on computational cost requirements.
* Description of the computational resources used for training MindEye and comparison of model size with other reconstruction approaches (see new Appendix A.10 in attached pdf). Notably, we mention how MindEye, despite having a very large parameter count, is actually quite computationally efficient due to the majority of parameters stemming from linear layers (MLPs are more computationally efficient than transformers or CNNs). MindEye can be fully trained on a single A100 in less than 18 hours.
* We elaborate on the theoretical motivation for aligning disjointed CLIP embeddings by referring to the “modality gap” geometric phenomenon. This describes the underlying reason behind why contrastive learning produces disjointed embeddings and thus motivates our use of a diffusion prior for alignment.

Because a few reviewers brought up the concern of novelty, below we summarize the novel advances put forth by our paper (:

1. Our novel implementation of separate submodules within a single model was shown to be critical for attaining simultaneous state-of-the-art reconstruction and retrieval metrics.
2. Contrary to common expectations within the neuroimaging community, using a deep MLP with a parameter count orders of magnitude higher than the number of training samples did not produce overfitting and instead benefitted model performance.
3. We introduce a novel bidirectional version of mixup contrastive data augmentation that seems to work very well in our low sample setting.
4. Mapping voxels to Stable Diffusion’s VAE latent space produces state-of-the-art image reconstructions in terms of low-level image metrics.
5. We are the first paper to attempt large-scale image retrieval (using LAION-5B) from fMRI brain data inputs.

---

### Decision · Program_Chairs · 2023-09-21

**Decision:**

Accept (spotlight)

**Comment:**

This paper showcases an exciting use of contrastive learning and diffusion to respectively identify the image seen by an individual from their brain activity and to reconstruct the image from the brain activity. While these two goals are not new, even to neurips, the current work has been found to both be cleanly implemented and have good qualitative and quantitative performance. The authors are expected to incorporate the new changes into their paper.